# SysMoBench: Evaluating AI on Formally Modeling Complex Real-World Systems

**Qian Cheng**[♠]   **Ruize Tang**[♡]   **Emilie Ma**[♣]   **Finn Hackett**[♣]   **Peiyang He**[♠]
**Yiming Su**[◇]   **Ivan Beschastnikh**[♣]   **Yu Huang**[♠][*]   **Xiaoxing Ma**[♠]   **Tianyin Xu**[◇]
[♠]State Key Lab of Novel Software Technology, Nanjing University, China,
{cq, peiyang_he}@smail.nju.edu.cn, {xxm, yuhuang}@nju.edu.cn
[♡]Microsoft Research Asia, ruizetang@microsoft.com
[♣]University of British Columbia, contact@emilie.ma, {fhackett, bestchai}@cs.ubc.ca
[◇]University of Illinois Urbana-Champaign, {yiming34, tyxu}@illinois.edu

## Abstract

Formal models are essential to specifying large, complex computer systems and verifying their correctness, but are notoriously expensive to write and maintain. Recent advances in generative AI show promise in generating certain forms of specifications. However, existing work mostly targets small programs, not complete systems. It is unclear whether AI can deal with realistic system artifacts, as this requires abstracting their complex behavioral properties into formal models. We present SysMoBench, a benchmark that evaluates AI's ability to formally model large, complex systems. We focus on concurrent and distributed systems, which are keystones of today's critical infrastructure, encompassing operating systems and cloud infrastructure. We focus on TLA$^+$, the *de facto* specification language for concurrent and distributed systems, though SysMoBench has been extended to other languages. We address the primary challenge of evaluating AI-generated models by automating metrics like syntactic and runtime correctness, conformance to system code, and invariant correctness. SysMoBench currently includes eleven diverse system artifacts: the Raft implementation of Etcd and Redis, ZooKeeper's leader election, the Spinlock, Mutex, and Ringbuffer in Asterinas OS, etc., with more being added. SysMoBench enables us to understand the capabilities and limitations of today's LLMs and agents, providing a firm footing for tools in this area and opening up promising new research directions.

## 1 Introduction

Formal models are essential to specifying computer systems and reasoning about their correctness. They provide a mathematical foundation to verify the *design* of complex systems, such as distributed protocols and concurrent algorithms (Lamport, 2002; Tasiran et al., 2003; Newcombe et al., 2015; Hackett et al., 2023b). Recently, formal models are used to describe system *implementations*—system code that runs on user devices and in production environments. Such *system models* enable verification of system code via comprehensive testing and model checking (Bornholt et al., 2021; Tang et al., 2024; Ouyang et al., 2025; Tang et al., 2025). For example, system models of ZooKeeper were used to detect deep bugs that violate system safety and verify their fixes (Ouyang et al., 2025).

However, system models are notoriously expensive to write and maintain. Different from protocols and algorithms, system code contains low-level details, is more complex, and constantly evolves. Hence, synthesis of system models is an open challenge (e.g., TLAi+ Challenge (2025)).

Recent advances in generative AI, represented by large language models (LLMs) and agentic techniques, show promise in generating function-level specifications, in the form of pre- and post-conditions (Rego et al., 2025; Cao et al., 2025; Xie et al., 2025; Chakraborty et al., 2025; Ma et al., 2025). This indicates that AI can capture certain behaviors of software programs. However, it is unclear whether AI could effectively model a complex system, which requires altogether different capabilities than the synthesis of pre- and post-conditions of a function. Modeling a system requires

---

[*]Corresponding author.

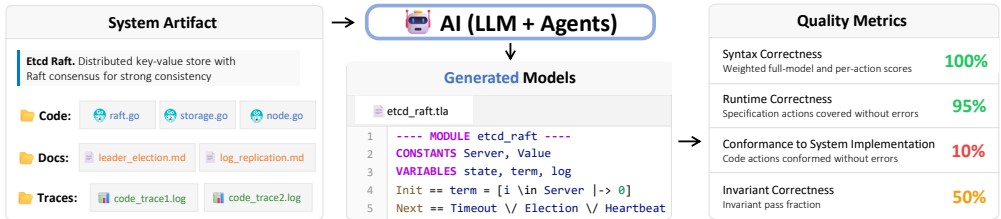

Figure 1: SYSMOBENCH sources its tasks from real-world systems (e.g., Etcd Raft in the figure). It automatically evaluates the system models in TLA$^+$ generated by AI with different metrics.

the AI to understand the system design (e.g., the underlying protocols and algorithms), reason about safety and liveness under unexpected faults and external events, and abstract system behavior into an executable program. It is unclear to what extent AI has such capabilities.

In this paper, we present SYSMOBENCH, a benchmark to evaluate AI's ability to formally model complex systems. We target all forms of generative AI, including LLMs and agentic techniques. We focus on *concurrent and distributed systems*, which are especially difficult to model. They also underpin critical modern infrastructure, like operating systems and cloud computing. We focus on TLA$^+$, the *de facto* formal specification language for concurrent and distributed systems (§2). SYS-MOBENCH can be easily extended to support other specification languages such as Alloy (Jackson, 2012), PAT (Sun et al., 2009), P (Desai et al., 2013), and SPIN (Holzmann, 1997). We have added support for Alloy and PAT (see Appendix B).

The key challenge of SYSMOBENCH is to *automatically* evaluate AI-generated models—how can we tell if a system model is of high quality? We did not find any directly applicable metrics in use by existing work on TLA$^+$ specification generation. For example, Cao et al. (2025) only checks if the generated TLA$^+$ can be run by the TLC model checker (Yu et al., 1999). But successfully running TLC is not an indicator of whether the model correctly describes the system. One approach is to evaluate AI-generated pre-/post-conditions (Rego et al., 2025; Ma et al., 2024) against human-written reference specifications. However, such a comparison can be brittle, and real-world systems rarely have such low-level specifications. Writing a system model remains a highly challenging task that requires months to years of expert effort.

A key contribution of our benchmark is quality metrics that can be *automatically* checked. These metrics reflect the fundamental requirements of a formal system model for use cases like formal verification (Lamport, 2002) and model-driven testing (Clarke et al., 2018).

- **Syntax correctness.** We statically check whether the generated system model uses valid TLA$^+$ syntax using the SANY Syntactic Analyzer.

- **Runtime correctness.** We check how much of the generated TLA$^+$ can be executed using the TLC model checker (Yu et al., 1999), which is a proxy for logical self-consistency.

- **Conformance.** We measure whether the model conforms to the system implementation via trace validation (Cirstea et al., 2024; Tang et al., 2025; Hackett & Beschastnikh, 2025).

- **Invariant correctness.** We model-check the system model against system-specific invariants that reflect the system's safety and liveness properties.

SYSMOBENCH currently includes eleven real-world artifacts, including distributed systems like Etcd, Redis, and ZooKeeper, and concurrent systems like spinlock, mutex, and ringbuffer from Asterinas OS. We also include system artifacts synthesized by PGo (Hackett et al., 2023a) to evaluate AI's ability to comprehend *generated* system code. More system artifacts are actively being added.

SYSMOBENCH enables us to understand the capabilities and limitations of AI in using TLA$^+$ to model real-world systems by evaluating different agent designs with various AI models. State-of-the-art LLMs show good performance in modeling small system artifacts such as a spinlock implementation. On the other hand, these LLMs show limited ability to comprehend and abstract large, complex systems such as a Raft implementation (Ongaro & Ousterhout, 2014). Overall, we believe that SYSMOBENCH can spur innovative AI approaches in the context of formal system models, similar to the role of SWE-bench (Jimenez et al., 2024) in software engineering.

SYSMOBENCH can be found here: `https://github.com/specula-org/SysMoBench`.

## 2  BACKGROUND

SYSMOBENCH focuses on formal models written in TLA$^+$ (Lamport, 2002), which is the *de facto* specification language for modeling distributed and concurrent systems in practice. TLA$^+$ is chosen from a practical standpoint, not a language standpoint (SYSMOBENCH supports other specification languages; Appendix B). TLA$^+$ is widely used by software companies like Amazon, Microsoft, Nvidia, Google, Oracle, etc. (see TLA+ Foundation (2025)) to verify critical infrastructure systems such as distributed consensus systems (e.g., Etcd and ZooKeeper), confidential consortium frameworks (Howard et al., 2025), databases (e.g., CosmosDB and MongoDB), OS kernel synchronization (Tang et al., 2025), and cache coherence (Beers, 2008).

A TLA$^+$ model specifies system behaviors as a collection of state variables, an initial predicate that defines their initial values, a next-state relation that determines state transitions, and temporal properties that specify correctness requirements. The next-state relation is expressed as multiple actions, each describing an atomic state update of all variables. TLA$^+$ is built upon the Temporal Logic of Actions (TLA), which *includes* and *extends* standard linear temporal logic (LTL) (Pnueli, 1977), providing a rigorous mathematical foundation for reasoning about system behavior over time. TLA$^+$ models can be verified using explicit-state model checking via TLC (Yu et al., 1999), symbolic model checking via Apalache (Konnov et al., 2019), and deductive verification via the TLA$^+$ Proof System (Chaudhuri et al., 2010). In SYSMOBENCH, we primarily use TLC, the most widely used TLA$^+$ tool that systematically explores all reachable states of a system model to ensure that properties hold over the entire state space. These characteristics make TLA$^+$ particularly well-suited for modeling complex concurrent and distributed systems.

```rust
1  pub struct SpinLock<T> { lock: AtomicBool }
2  pub struct SpinLockGuard<T, R: SpinLock<T>> {
3    guard: R,
4  }
5  impl<T> SpinLock<T> {
6    pub const fn new(val: T) -> Self {
7      SpinLock { lock: AtomicBool::new(false) }
8    }
9    pub fn lock(&self) -> SpinLockGuard<T> {
10     self.acquire_lock();
11     SpinLockGuard { guard: self }
12   }
13   fn acquire_lock(&self) {
14     while !self.try_acquire_lock() {}
15   }
16   fn try_acquire_lock(&self) -> bool {
17     self.lock.compare_exchange(false, true)
18       .is_ok()
19   }
20   fn release_lock(&self) { // on guard drop
21     self.lock.store(false);
22   }
23 }
```

```
1  CONSTANTS Threads
2  VARIABLES lock_state, pc
3  Init ==
4    lock_state = FALSE /\ pc = [t \in Threads |-> "idle"]
5  StartLock(t) ==
6    /\ pc[t] = "idle"
7    /\ pc' = [pc EXCEPT ![t] = "trying_blocking"]
8    /\ lock_state' = lock_state
9  Acquire(t) ==
10   /\ pc[t] \in {"trying_blocking", "spinning"}
11   /\ IF lock_state = FALSE
12      THEN /\ lock_state' = TRUE
13           /\ pc' = [pc EXCEPT ![t] = "locked"]
14      ELSE /\ pc' = [pc EXCEPT ![t] = "spinning"]
15           /\ lock_state' = lock_state
16 Unlock(t) ==
17   /\ pc[t] = "locked"
18   /\ lock_state' = FALSE
19   /\ pc' = [pc EXCEPT ![t] = "idle"]
20 Next == \E t \in Threads:
21   StartLock(t) \/ Acquire(t) \/ Unlock(t)
22 MutualExclusion ==
23   Cardinality({t \in Threads : pc[t] = "locked"}) <= 1
```

Figure 2: Simplified code that implements a spinlock in Asterinas (left) and an AI-generated TLA$^+$ model (right). This spinlock represents one of the simplest systems in SYSMOBENCH.

Figure 2 shows simplified code that implements a spinlock in the Asterinas operating system (Peng et al., 2025) and the corresponding TLA$^+$ model that describes the code. The TLA$^+$ model is generated by the AI agent we evaluate in §5. The model defines constants such as Threads (line 1) and system-state variables such as lock_state and pc (line 2). The initial state Init (line 3) assigns initial values to all variables. Three actions are defined (lines 5–19)—StartLock, Acquire, and Unlock—corresponding to the code logic, where Acquire combines the logic of acquire_lock and try_acquire_lock. Each action is enabled by certain conditions, e.g., StartLock is enabled when a thread's pc is "idle"; it then assigns next-state values to all variables.

To model the spinlock implementation, the AI must first understand the behavior of each function. Next, it must decide how to represent the system. This involves introducing variables, such as auxiliary ones like pc, and defining atomic actions that preserve concurrency semantics. Finally, the AI must specify correctness properties. For example, mutual exclusion (line 22) requires that in every state, at most one thread can be in the "locked" state.

Note that SYSMOBENCH concerns formal models of system implementations, or *system specifications* in the TLA$^+$ and formal method literature. A system model enables verification of system code, but does not necessarily capture design requirements (Stoica et al., 2024). SYSMOBENCH does not target other forms of specifications, such as formal proofs (Chen et al., 2025) or function-level pre- and post-conditions (Ma et al., 2025).

## 3 SYSMOBENCH

SYSMOBENCH is a benchmark that uses real-world distributed and concurrent system artifacts to evaluate AI's ability to formally model systems. Table 1 lists the systems that have been integrated in SYSMOBENCH; we are actively adding more system artifacts (§3.3).

Table 1: System artifacts that have been integrated in the SYSMOBENCH; "TLA$^+$ LoC" refers to the AI-generated TLA$^+$ models presented in our evaluation results (§5).

| System | Type | Desc. | Source Lang. | Source LoC | TLA$^+$ LoC |
|---|---|---|---|---|---|
| Asterinas Spinlock | Concurrent | Synchronization | Rust | 213 | 151 |
| Asterinas Mutex | Concurrent | Synchronization | Rust | 186 | 219 |
| Asterinas Rwmutex | Concurrent | Synchronization | Rust | 395 | 250 |
| Asterinas Ringbuffer | Concurrent | Data Structure | Rust | 615 | 123 |
| Etcd Raft | Distributed | Consensus (Raft) | Go | 2,159 | 385 |
| Redis Raft | Distributed | Consensus (Raft) | C | 2,394 | 349 |
| Xline CURP | Distributed | Replication (CURP) | Rust | 4,064 | 100 |
| ZooKeeper FLE | Distributed | Leader Election | Java | 5,360 | 141 |
| PGo dqueue | Distributed | Distributed Queue | Go | 175 | 75 |
| PGo locksvc | Distributed | Lock Server | Go | 281 | 93 |
| PGo raftkvs | Distributed | Consensus (Raft) | Go | 3,163 | 508 |

### 3.1 TASK FORMULATION

Each SYSMOBENCH task comprises generating a system model for a given system artifact (Table 1). SYSMOBENCH does not concern how the system model is generated. It can be generated by prompting LLMs directly, with few-shot learning, or with agentic techniques that invoke external tools (we evaluate both in §5). Since system artifacts in SYSMOBENCH are real-world system projects, one can feed various data sources to the LLMs/agents, such as source code, documents, and runtime traces. This task mirrors real-world modeling workflows of human engineers.

Each task specifies the granularities at which to model the target system's essential behavioral properties and state transitions. The required level of granularity is defined based on target use cases; our current use case is model-checking based system verification, so we require the same level of detail as in prior verification and bug finding work. The model must include core actions that interact with other components, while excluding implementation details unrelated to system behavior. We evaluate behavioral conformance rather than structural equivalence, allowing fine-grained modeling of core actions as long as they preserve semantic obligations needed for verification. Each task concretely lists core actions that must be modeled and actions that should be excluded. Take Spinlock as an example (Figure 2): the requirements are specified as follows:

```
Mandatory core actions that must be modeled:
• The model must specify lock() and unlock() actions.
• Atomic compare_exchange operation on the lock variable.
• Spinning when the lock is contended.

Actions that should be excluded from the model:
• RAII guard implementation details.
• Non-core details (e.g., debug formatting and trait implementation).
```

Besides, the task also requires generating a TLC configuration as a part of the model.

## 3.2 Metrics and Their Measurement

Key contributions of SYSMOBENCH are to (1) define fair metrics for the quality of AI-generated TLA$^+$ models, and (2) design practical techniques to automate metric measurements. SYSMOBENCH does not rely on human evaluation, which is hard to scale, especially for complex systems. We do not consider LLM-as-a-judge approaches, as we find these unreliable and uninterpretable.

SYSMOBENCH includes four metrics that evaluate a TLA$^+$ model on syntax (§3.2.1), runtime correctness (§3.2.2), conformance to system code (§3.2.3), and invariant correctness (§3.2.4). The metrics are not independent, e.g., a model with syntax errors cannot be evaluated for other metrics. An executable model is evaluated for both conformance and invariant correctness. We design *partial scoring* schemes for every metric and normalize results to percentage values, making them easy to interpret. Figure 3 illustrates the metrics and the evaluation workflow.

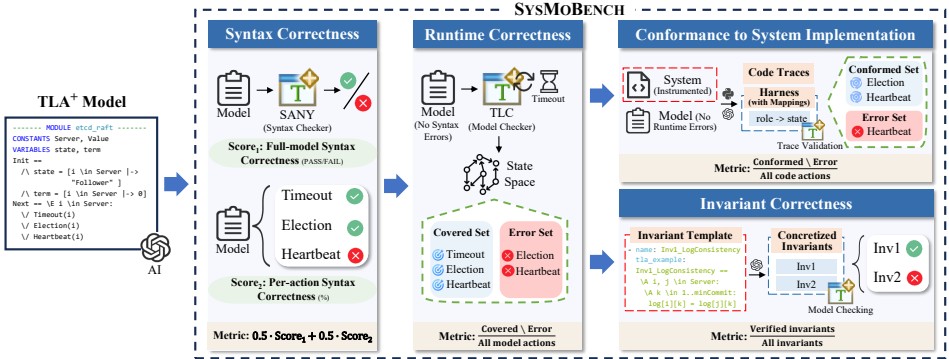

Figure 3: Metrics and evaluation workflow of SYSMOBENCH. The red dashed boxes denote inputs provided by the system artifact: instrumented system for code traces and required invariants.

### 3.2.1 Syntax Correctness

SYSMOBENCH uses the TLA$^+$ SANY Syntactic Analyzer (Lamport, 2002) to check the models' syntax against TLA$^+$ grammar rules, operator usage, module structure, etc. Note that SANY checks the entire specification. If the model passes the SANY checks, it earns a full score. However, many AI-generated models fail; we thus need fine-grained analysis for partial scoring.

SYSMOBENCH offers per-action analysis for partial scoring by checking how many generated actions are erroneous (and failed SANY). It encapsulates each action in the model into a per-action model by adding necessary dependencies (e.g., constant declarations, variable definitions, etc.). It then uses SANY to check the syntax of per-action model (only syntax correction is concerned in this step; no equivalence check). A partial score $S$ represents the percentage of correct actions $n_c$ among the total actions $n_t$, i.e., $S = \frac{n_c}{n_t}$. Here, $n_c$ is determined by running SANY on each per-action module and counting those that pass without syntax errors, while $n_t$ is obtained by counting all action definitions in the original model. Note that the per-action checks do not account for inter-action dependencies: a model that passes all the per-action checks can still fail. We use a weighted scoring scheme that gives equal weights to per-action correctness and inter-action correctness. A model that passes the overall SANY check earns 100%, while passing only all per-action checks earns 50%. Only system models with 100% syntax scores will be evaluated for other metrics, because models with syntax errors cannot be compiled or executed (which is required by other metrics).

### 3.2.2 Runtime Correctness

For a syntactically correct system model, SYSMOBENCH next evaluates if the model can be executed correctly. To do so, SYSMOBENCH performs bounded model checking and simulation using TLC, and then observes covered actions and runtime errors (if any) by parsing TLC's coverage report and error output. This model checking and simulation explores the state space without any invariant checking (see §3.2.4). During this state space exploration, SYSMOBENCH records all covered actions and the actions with runtime errors.

We define a metric $M_r$ that represents the coverage of actions without runtime errors: $M_r = \frac{n_r}{n_t}$, where $n_r$ is the number of covered actions that did not report errors during state exploration, and $n_t$ is the total number of actions in the model. Models with no runtime errors can then be executed to explore state space. Only such models are evaluated for conformance and invariant correctness.

### 3.2.3 CONFORMANCE TO SYSTEM IMPLEMENTATION

SYSMOBENCH evaluates an executable model's conformance to the system implementation using trace validation (Cirstea et al., 2024). Trace validation checks whether system execution traces correspond to paths in the model's state space. SYSMOBENCH supports diverse trace validation mechanisms (Tang et al., 2025; Cirstea et al., 2024; Hackett & Beschastnikh, 2025).

Specifically, to collect execution traces, system code is instrumented with logging statements. The instrumentation granularity matches the granularity requirements of the task. If the AI-generated model is coarser than the trace logs, conformance checking may fail; otherwise, we use missing-event inference techniques (Tang et al., 2025) to account for uninstrumented actions.

The key challenge of automatic conformance checking of any AI-generated models is to correctly map the elements in the model to elements in the system execution log. This is because AI will often use names that differ from those in system code. We solve this problem by using a coding LLM (e.g., Claude-Sonnet-4) to (1) extract constants, variables, and actions from the input model and (2) map them to the corresponding elements specified in the task requirement (§3.1).

The use of LLMs to automatically map elements in the model and code may raise reliability concerns. In our experience, state-of-the-art LLMs accomplish this mapping task reliably (§4). This is because (1) the mapping task is simple and well-defined; (2) the generated models are derived from the system artifacts and thus largely follow the system's naming conventions; and, (3) our trace validation technique (Tang et al., 2025) can tolerate a certain level of missing variables or actions, though we have not found such cases so far. A similar use of LLMs for mapping is adopted in TLAi+Bench (2025) (discussed in §6).

During trace validation across all traces, SYSMOBENCH keeps track of code actions that are covered and those that trigger errors. Specifically, SYSMOBENCH feeds the trace to TLC along with the model and records whether TLC successfully validates the trace. If validation fails, SYSMOBENCH identifies where the mismatch occurred by analyzing TLC's trace validation output. To measure conformance, we define $M_c$ as the coverage of code actions without conformance errors: $M_c = \frac{n_c}{n_t}$, where $n_c$ is the number of code actions that were covered during validation with no errors, and $n_t$ is the total number of actions in the instrumented code. We use instrumented code actions instead of model actions because this provides a stable, implementation-grounded granularity that is consistent across different AI-generated models.

### 3.2.4 INVARIANT CORRECTNESS

SYSMOBENCH also evaluates whether AI-generated models always satisfy invariants that describe the expected safety and liveness properties of the system. In principle, if a system model fully conforms to code, violations of these invariants would indicate bugs in system code; in practice, few AI-generated models achieved fine-grained conformance. Nevertheless, AI-generated models have demonstrated practical utility by successfully reproducing known bugs from previous system versions (Appendix C.2). Table 2 lists the invariants for the spinlock code in Figure 2. These invariants are part of the benchmark defined by the task (§3.3).

Table 2: Example spinlock invariants

| Invariants | Description | Type |
| --- | --- | --- |
| Mutual exclusion | At most one process can be in the critical section at any time | Safety |
| Lock consistency | The lock state accurately reflects critical section occupancy | Safety |
| No deadlock | Not all threads can be stuck spinning simultaneously | Safety |
| Guard lifecycle | Every thread eventually releases the lock it acquires | Liveness |
| Eventual release | The system eventually reaches a state where all threads are idle | Liveness |

SYSMOBENCH addresses a similar challenge as in §3.2.3: it needs to automatically map the actions, variables, and data structures in the system model to those expressed in the invariants. For this,

the invariants in SYSMOBENCH are templates that contain a description of the property, formal definitions, and example TLA$^+$ invariants. We then use an LLM to translate these templates into model-specific invariants that can be checked against the system model. For example, the following template defines the mutual exclusion invariant in Table 2:

```
- name:"MutualExclusion"
  type:"safety"
  natural_language:"Only one thread can access a shared resource at a time"
  formal_description:"No more than one thread in the critical section"
  tla_example:MutualExclusion == Cardinality({t \in Threads:pc[t] = "in_cs"}) <= 1
```

SYSMOBENCH prompts the LLM with both the invariant template and the system model and asks it to concretize the template using the model. This mapping is highly structured: the output substitutes the template's variables and constants with those in the model. For example, the mutual exclusion invariant, `Cardinality({t \in Threads: status[t] = "locked"}) <= 1`, is a concretization of the template by replacing `pc` with `status` and `in_cs` with `locked`. We evaluate the reliability of this LLM-assisted concretization in §4.

The invariants are used by TLC during model checking, and SYSMOBENCH observes whether each invariant is violated. Specifically, for each invariant, SYSMOBENCH creates a separate model with that invariant and runs TLC independently. This allows SYSMOBENCH to record whether each invariant is violated. We define a metric $M_i$ that represents the *fraction of invariants passed*, denoted as $M_i = \frac{n_i}{n_t}$, where $n_i$ is the number of invariants that hold across the explored state space, and $n_t$ is the total number of invariants defined for the model. Models with a higher $M_i$ are of higher quality. When combined with runtime and conformance coverage metrics, a higher $M_i$ increases confidence in the correctness of the input specification.

### 3.3 ADDING NEW SYSTEMS AND SPECIFICATION LANGUAGES TO SYSMOBENCH

SYSMOBENCH provides an extensible framework to add more real-world system artifacts. To add a new artifact to SYSMOBENCH, one needs to (1) prepare the system artifact (e.g., source code and documents); (2) create a new task that specifies the abstractions and components to model (§3.1); (3) develop an invariant template (§3.2.4) that specifies correctness properties (safety and liveness); and (4) provide a harness for trace validation by instrumenting system code. In our experience, the effort to add a new system artifact to SYSMOBENCH is manageable. For example, adding Etcd Raft took one SYSMOBENCH author four days; an Xline CURP contributor with no experience of SYSMOBENCH added the system to SYSMOBENCH in four days. Most of the effort is spent on instrumenting the system to collect execution logs for trace validation in order to measure conformance. Unlike other benchmarks (§6), SYSMOBENCH does not require writing reference models; in fact, we hope that some of the AI-generated models can eventually be adopted by real-world system projects.

SYSMOBENCH is extensible to formal specification languages other than TLA$^+$. We extended SYSMOBENCH to support Alloy (Jackson, 2012) and PAT (Sun et al., 2009), demonstrating its generality. Details of these extensions and preliminary evaluation are presented in Appendix B. The results show that while our framework is extensible, TLA$^+$ remains the practical choice and can benefit from AI-driven techniques (existing LLMs are less familiar with Alloy and PAT).

## 4 EVALUATION SETUP

To evaluate AI's system modeling abilities, we use three agents powered by LLMs.

- **Basic Modeling Agent.** This agent reflects the LLM's raw modeling abilities. The agent prompts an LLM with the system source code and the task requirement (§3.1) (see Appendix G).

- **Code Translation Agent.** This agent uses an LLM to *translate* system code into equivalent TLA$^+$. The agent translates code statement by statement (from the source language to TLA$^+$), and then organizes these statements' control flow into a TLA$^+$ model. The agent reflects LLM-based code translation capabilities. We adopt Specula (2025) as our code translation agent.

- **Trace Learning Agent.** This agent does not use code as input, but tries to learn the system model from system traces. It prompts LLMs with the traces to infer the system model (see Appendix H). This agent reflects the capability of automata learning (Biermann & Feldman, 1972) with LLMs.

We follow HumanEval (Chen et al., 2021): we run each agent five times and evaluate the best output model. The agents can enhance the model via feedback loops (max. three iterations) if the generated model cannot pass compilation or has runtime errors. No human intervention is allowed.

We use four different LLMs to power the three agents: Claude-Sonnet-4 (20250514), GPT-5 (20250807), Gemini-2.5-Pro (20250617), and DeepSeek-R1 (20250528). We run the SANY Syntactic Analyzer, TLC model checker, and system code (for conformance checking) on a server with dual AMD EPYC 7642 48-Core Processors and 256GB RAM running Ubuntu 22.04.

**Robustness of LLM-assisted Components.** SYSMOBENCH uses LLM-assisted techniques to map elements in an AI-generated TLA$^+$ model to those in the system logs (§3.2.3), and to concretize invariant templates (§3.2.4). We inspected the LLM-assisted mapping and concretization, and found the results to be correct. We also conducted an experiment using the "gold model" for Etcd Raft and Asterinas spinlock, which are known to be correct. We created 10 models (5 for each system) by changing the names of the variables and actions and tweaking the model's granularity. The gold models achieve a perfect score on all metrics, empirically validating the quality of our metrics.

**Training Data Contamination.** One may be concerned about the fairness of SYSMOBENCH because it uses open-source projects where the system code likely already appears in LLM training data. In fact, having system code in LLM training data is intended. The design mirrors how human engineers write formal models: they first learn system code before writing formal models. Our goal is to leverage LLMs to write effective TLA$^+$ models for important, safety-critical software systems, which requires LLMs to have internalized knowledge of these systems. Note that this is different from coding benchmarks in that we ask LLMs/agents to write existing code.

Second, few system artifacts in SYSMOBENCH have TLA$^+$ system models in their open-source repositories. The TLA$^+$ models of Asterinas Spinlock/Mutex/Rwmutex are never released. Redis Raft and Xline CURP do not have any TLA+ models. Etcd Raft and PGo systems do have TLA+ models in the repositories. However, those models are for protocols, not for system code.

## 5 RESULTS

We present evaluation results for the *basic modeling agent* and the *code translation agent* on Asterinas Spinlock and Etcd Raft (Table 3). Appendix I.2 contains the complete results for all systems in SYSMOBENCH. We omit the results of the trace learning agent (which fails to pass runtime checks).

Table 3: Evaluation results of two AI agents on two representative system artifacts. ✓ and ✗ mark whether the model is evaluated in the next phase of measurements (see Figure 3).

(a) Asterinas Spinlock

| Agent | LLM | Syntax | Runtime | Conformance | Invariant |
|---|---|---|---|---|---|
| Basic Modeling | Claude-Sonnet-4 | 100.00% ✓ | 100.00% ✓ | 100.00% | 100.00% |
| | GPT-5 | 100.00% ✓ | 100.00% ✓ | 80.00% | 100.00% |
| | Gemini-2.5-Pro | 100.00% ✓ | 100.00% ✓ | 80.00% | 85.71% |
| | DeepSeek-R1 | 100.00% ✓ | 100.00% ✓ | 80.00% | 100.00% |
| Code Translation | Claude-Sonnet-4 | 100.00% ✓ | 100.00% ✓ | 100.00% | 100.00% |
| | GPT-5 | 100.00% ✓ | 100.00% ✓ | 100.00% | 85.71% |
| | Gemini-2.5-Pro | 100.00% ✓ | 100.00% ✓ | 100.00% | 100.00% |
| | DeepSeek-R1 | 100.00% ✓ | 100.00% ✓ | 100.00% | 100.00% |

(b) Etcd Raft

| Agent | LLM | Syntax | Runtime | Conformance | Invariant |
|---|---|---|---|---|---|
| Basic Modeling | Claude-Sonnet-4 | 100.00% ✓ | 25.00% ✓ | 7.69% | 69.23% |
| | GPT-5 | 47.87% ✗ | - | - | - |
| | Gemini-2.5-Pro | 50.00% ✗ | - | - | - |
| | DeepSeek-R1 | 50.00% ✗ | - | - | - |
| Code Translation | Claude-Sonnet-4 | 100.00% ✓ | 66.67% ✓ | 15.38% | 92.31% |
| | GPT-5 | 100.00% ✓ | 20.00% ✗ | - | - |
| | Gemini-2.5-Pro | 44.44% ✗ | - | - | - |
| | DeepSeek-R1 | 100.00% ✓ | 0.00% ✗ | - | - |

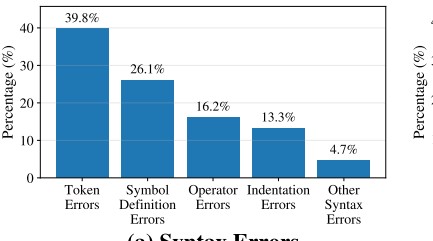 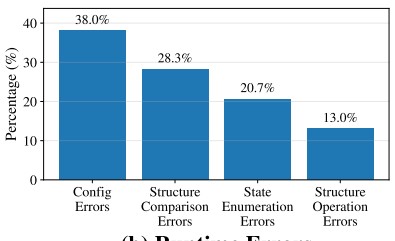 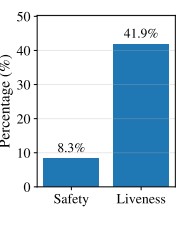

(a) Syntax Errors       (b) Runtime Errors       (c) Invariant Violations

Figure 4: LLM error attribution regarding the SYSMOBENCH metrics in the basic modeling agent. The conformance metric is omitted as it has a single attribution.

**Modeling Capability.** We focus on the results of the basic modeling agent. The basic modeling agent can generate high-quality TLA$^+$ models for Spinlock, which is among the simplest artifacts in SYSMOBENCH (Table 1), showing certain levels of modeling capability. However, for larger and more complex systems such as the distributed protocol implementations, the basic modeling agent performs poorly. For Etcd Raft, only with Claude-Sonnet-4, the modeling agent reaches the conformance and invariant checking, and scores are low. Clearly, the complexity and size of Etcd Raft exceed the modeling ability of the LLMs and agents.

For Etcd Raft, the basic modeling agents struggle with (1) code verbosity, (2) protocol complexity, and (3) abstraction. Etcd Raft has much more code than Spinlock, with low-level utilities (e.g., for debugging) and implementation-specific comments, which often cause agents to lose focus on essential system logic. Moreover, the Raft protocol (Ongaro & Ousterhout, 2014) has more complex logic than a spinlock in terms of ordering and intricate conditions of state transitions. Both (1) and (2) make Etcd Raft significantly more challenging for LLMs to comprehend the system artifact. For (3), Etcd Raft presents significant abstraction challenges: concepts like distributed logs require nested data structures, demanding LLMs to precisely express them using TLA$^+$ language constructs.

The basic modeling agents also perform poorly on PGo systems (Appendix I.2), indicating limited LLM ability to comprehend machine-generated systems. Code in PGo-generated systems is a mix of compiler-generated patterns and a runtime library (Appendix E). The generated code is repetitive, and, while it borrows some variable names from the source specification, intermediate variables have synthetic, non-significant names, which provide few semantic clues to an LLM (or a human reader).

**Analysis on Agents.** For complex systems like Etcd Raft, the code translation agent outperforms the basic modeling agent. We believe this is due to powerful LLM translation abilities (Yang et al., 2024b). This approach allows us to leverage symbolic control-flow analysis for rigorous synthesis and prevent logic hallucinations by adhering to system code. We also saw that LLMs would sometimes imitate classic TLA$^+$ models from their training set, missing important system-specific content.

**Analysis on Invariants.** LLMs violate very different types of invariants—only 8.3% of safety properties are violated while 41.9% of liveness properties were violated (Figure 4c). This indicates limited LLM ability to reason temporally. To understand these violations, we conducted a fine-grained analysis categorizing them by root causes (Appendix I.1). We find that while fairness assumption violations (e.g., missing or incorrectly specified assumptions) are a significant issue across systems, logical and structural errors tend to manifest earlier before fairness-related issues emerge.

**Analysis on LLMs.** We observe that LLMs constantly introduce syntax errors (Figure 4a), especially GPT-5, Gemini-2.5-Pro, and DeepSeek-R1. For example, DeepSeek-R1 often misuses mathematical symbols (e.g., $\cap$, $\forall$) instead of ASCII TLA$^+$ operators. Gemini-2.5-Pro and GPT-5 tend to mix up TLA$^+$ syntax with other languages like Python. LLMs also provide incorrect operator parameters and produce malformed indentation. In terms of runtime errors (Figure 4b), LLMs frequently generate inconsistent TLC configs, with missing constants or mismatched declarations. Misunderstanding TLA$^+$ data structures is also common, e.g., comparing incompatible types.

Claude-Sonnet-4 outperforms other evaluated LLMs in most metrics across evaluated system artifacts. Since only syntax-valid models can proceed to subsequent evaluation phases, Claude-Sonnet-4's ability to generate syntactically correct TLA$^+$ models provides an initial advantage. However,

its strength extends beyond syntax correctness. SYSMOBENCH decomposes evaluation into four distinct metrics that separate syntactic correctness from system behavior reasoning. As shown in Appendix I.2, for models that pass syntax checks, Claude-Sonnet-4 generally still achieves higher scores on runtime, conformance, and invariant metrics compared to other LLMs.

**Qualitative Assessment.** We conducted qualitative evaluation to assess AI-generated system model quality and utility in terms of bug finding (Appendix C). Comparing with human-written TLA$^+$ models, AI-generated TLA$^+$ models differ in structure and completeness but they capture essential system behaviors. Despite these limitations, AI-generated models have successfully reproduced known bugs in five systems, demonstrating their practical utility for partial correctness checking.

## 6 RELATED WORK

SYSMOBENCH is the first framework that evaluates AI on formally modeling real-world systems.

**Benchmarks for Formal Specifications.** There are several benchmarks for evaluating LLMs and AI agents on generating function-level pre-/post-conditions and loop invariants (Rego et al., 2025; Xie et al., 2025; Cao et al., 2025; Chakraborty et al., 2025; Ma et al., 2025; Wen et al., 2024). These typically use small programs, such as VeriFast data structure specs (Rego et al., 2025) and Leet-Code programs (Ma et al., 2025). There also exist benchmarks for proof generation for deductive software verification (Yang et al., 2024a) and for verified code generation (Thakur et al., 2025; Ye et al., 2025). None of these benchmarks target real-world systems as in SYSMOBENCH. They fundamentally evaluate AI's abilities of code comprehension and specification, not system modeling. Similarly, PAT-Agent (Zuo et al., 2025) and Alloy-APR (Alhanahnah et al., 2025) target small tasks like puzzles and repairing injected errors (see Appendix B.3). The arguably most related benchmark is TLAi+Bench (2025) which evaluates AI-generated TLA$^+$ specifications, primarily logic puzzles. It evaluates AI's ability to *use* the TLA$^+$ language, not system comprehension or modeling. Hence, TLAiBench and related benchmarks like Cao et al. (2025) only measure the syntax and runtime correctness of the TLA$^+$ specifications. Li et al. (2025) develop a benchmark for Hyperkernel system call inference; however, it does not consider distributed systems or concurrency and assumes a ground-truth specification. As AI for code matures, the next step is capturing AI's benefits for practically verifying real-world systems. We developed SYSMOBENCH with this motivation in mind.

Our evaluation aims to establish a baseline of simple agents to reflect the status quo of AI abilities. More advanced agents, like those with domain-specific techniques such as Bhatia et al. (2024); Wang et al. (2025), can be developed to improve the quality of AI-generated models.

**General AI Benchmarks.** SYSMOBENCH differs from general AI reasoning benchmarks such as MMLU (Hendrycks et al., 2021), ARC (Clark et al., 2018), and HELM (Liang et al., 2022). These benchmarks evaluate generic reasoning, knowledge, and problem-solving capabilities across diverse domains, while SYSMOBENCH focuses on the specific task of formally modeling large, complex software systems as a foundation of formal system verification. SYSMOBENCH also differs from benchmarks targeting AI agent safety such as Agent-SafetyBench (Zhang et al., 2024). It currently targets traditional distributed and concurrent systems that are implemented in system code without neural components. The formal system modeling tasks evaluated by SYSMOBENCH are not covered by existing benchmarks such as EvalScope (EvalScope, 2024).

## 7 CONCLUDING REMARKS

This paper presents SYSMOBENCH, a new benchmark for evaluating generative AI in formally modeling real-world systems. This goal pushed us to articulate criteria for formal system models and to develop automatable metrics. We find that despite strong coding and debugging abilities, modern AI remains limited in comprehending and specifying complex systems. We hope SYSMOBENCH will help advance AI technologies towards software system intelligence, rather than code intelligence.

We are actively adding new system artifacts and improving SYSMOBENCH's usability. We encourage others to contribute their system artifacts. We are also exploring ways to measure the maintainability of AI-generated system models and considering ways to include human evaluation.

ACKNOWLEDGMENTS

We thank the TLAi+ Challenge organized by the TLA+ community, which started this joint effort. Huang's group is supported by the Frontier Technologies R&D Program of Jiangsu (BF2024059), the National Natural Science Foundation of China (62372222, 62025202), the CCF-Huawei Populus Grove Fund (CCF-HuaweiFM202505), and the Collaborative Innovation Center of Novel Software Technology and Industrialization. Beschastnikh's group is supported by the Natural Sciences and Engineering Research Council of Canada (NSERC), grant number RGPIN-2025-06826. Finn Hackett was partly funded by the MongoDB PhD Fellowship Program.

ETHICS STATEMENT

We strictly obeyed the principles outlined in the ICLR Code of Ethics, and carefully examined potential ethical concerns, including potential impacts on human subjects, practices to data set releases, potentially harmful insights, methodologies and applications, potential conflicts of interest and sponsorship, discrimination/bias/fairness concerns, privacy and security issues, legal compliance, and research integrity issues. We do not identify any potential risks. In fact, we believe that the work, together with its artifacts (e.g., the TLA+ models) will have positive impacts on the correctness of real-world computing systems and infrastructures.

REPRODUCIBILITY STATEMENT

We have made faithful efforts to ensure the reproducibility of our work. We have provided the details of our work in the paper and its appendix, including the prompts, implementations, and complete results. We have open-sourced all the research artifacts described in this paper at `https://github.com/specula-org/SysMoBench` which documents how to use and extend different parts of the benchmark. We expect that readers can easily reproduce our results reported in the paper. We also maintain an active forum to assist with reproduction problems and questions on how to use and build on SYSMOBENCH.

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

# A  ALTERNATIVE METRICS

No metric is perfect. Besides the core metrics presented in the paper, SYSMOBENCH also measures complementary metrics that provide different measures of the system model quality.

## A.1  RUNTIME PASS RATE

SYSMOBENCH repeatedly runs an agent to generate multiple TLA$^+$ system models and evaluates whether each system model passes the runtime checks. The runtime pass rate is defined as $M_{ar} = \frac{n_{ar}}{n_{at}}$, where $n_{ar}$ is the number of TLA$^+$ models that passed runtime checking, and $n_{at}$ is the total number of generated TLA$^+$ models. This metric complements the system model's action-level coverage metric $M_r$ (see §3.2.2), as it reflects the agent's ability and reliability to produce fully executable TLA$^+$ models. Note that a high $M_{ar}$ does not necessarily mean most actions in the TLA$^+$ models are correct. Even if some actions may contain runtime errors but are never executed during execution, the TLA$^+$ model can still pass runtime checking. Conversely, a low $M_{ar}$ may result from a small number of frequently failing actions rather than errors affecting many actions.

## A.2  CONFORMANCE PASS RATE

SYSMOBENCH repeatedly executes the system code to generate multiple code traces and checks which traces fully pass conformance checking. The conformance pass rate is defined as $M_{ac} = \frac{n_{ac}}{n_{at}}$, where $n_{ac}$ is the number of traces that passed conformance checking, and $n_{at}$ is the total number of traces generated. This metric complements the code action-level coverage metric $M_c$ (see §3.2.3) and provides a coarse-grained empirical measure of the TLA$^+$ model's overall alignment with observed system behavior. As with runtime correctness, a low $M_{ac}$ does not necessarily indicate that most actions are unconformed, while a high $M_{ac}$ generally suggests better overall system model quality, given sufficiently diverse traces.

# B  EXTENSIBILITY TO OTHER SPECIFICATION LANGUAGES

SYSMOBENCH is general to specification languages beyond TLA$^+$. To demonstrate its extensibility, we extended SYSMOBENCH to support Alloy (Jackson, 2012) and PAT (Sun et al., 2009).

## B.1  SUPPORTING PAT AND ALLOY

**PAT.** PAT (Process Analysis Toolkit) is a formal verification framework for concurrent and real-time systems. Supporting PAT in SYSMOBENCH is straightforward because PAT's tooling provides a workflow similar to TLA$^+$. We leverage PAT's parser for syntax checking, its simulator for runtime evaluation, and its assertion mechanism with model checking for invariant validation. Conformance is evaluated using PAT's native trace refinement checker. We implement adaptors to translate our concrete system traces into the PAT format, which are then validated against the PAT models.

**Alloy.** Alloy is a declarative specification language based on first-order relational logic. For Alloy support, evaluating syntax, runtime, and invariant correctness is straightforward using the Alloy Analyzer tool. Since Alloy does not provide a built-in notion of "action" as in TLA$^+$, we adapt the runtime metric by computing the proportion of variables and fields that become instantiated during bounded execution. This metric is analogous to the action-trigger coverage in TLA$^+$, and it indicates whether a model executes normally and whether certain branches are unreachable. For the conformance metric, we express a concrete system trace into Alloy facts, which are global constraints over a bounded sequence of states and must hold in all generated instances, for trace validation.

## B.2  EVALUATION RESULTS

We evaluated the basic modeling agent with four LLMs (Claude-Sonnet-4, GPT-5, Gemini-2.5-Pro, and DeepSeek-R1) on generating Alloy and PAT models for the Spinlock system, with three attempts per LLM. Table 4 shows the results. For both PAT and Alloy, the four evaluation metrics (syntax, runtime, conformance, and invariant) remain applicable. However, due to limitations of current tools, syntax checking for PAT and Alloy does not yet support partial scoring as in TLA$^+$.

Table 4: Preliminary results of Alloy and PAT support on Asterinas Spinlock using the basic modeling agent (3 attempts per LLM).

| Language | LLM | Syntax | Runtime | Conformance | Invariant |
|---|---|---|---|---|---|
| Alloy | Claude-Sonnet-4 | 0.00% | 0.00% | 0.00% | 0.00% |
| | GPT-5 | 100.00% | 0.00% | 0.00% | 0.00% |
| | Gemini-2.5-Pro | 0.00% | 0.00% | 0.00% | 0.00% |
| | DeepSeek-R1 | 0.00% | 0.00% | 0.00% | 0.00% |
| PAT | Claude-Sonnet-4 | 0.00% | 0.00% | 0.00% | 0.00% |
| | GPT-5 | 0.00% | 0.00% | 0.00% | 0.00% |
| | Gemini-2.5-Pro | 0.00% | 0.00% | 0.00% | 0.00% |
| | DeepSeek-R1 | 0.00% | 0.00% | 0.00% | 0.00% |

The AI-generated Alloy and PAT models are poor compared to TLA$^+$ models. For Alloy, only GPT-5 was able to generate a model that passes the syntax correctness check after multiple attempts, but the generated model scored 0% on runtime correctness. For PAT, none of the evaluated LLMs demonstrated familiarity with the PAT syntax—all generated PAT models failed syntax checks.

Our analysis reveals that current LLMs are unfamiliar with the syntax of Alloy and PAT. In practice, nearly all generated models failed at the parsing or type-checking stage. For PAT, LLMs frequently produced syntax borrowed from other languages such as C, Promela, or PRISM. For example, channels were often declared using PRISM-style range expressions (e.g., `channel acquire:{0..2};`) which PAT does not support. We also observed the introduction of keywords and type annotations that do not exist in PAT, such as adding explicit types (`int`) after variables or using `chan` instead of PAT's actual channel declaration syntax. For Alloy, we observed similarly systematic breakdowns. A common pattern was referencing signatures (types) that were never declared in the model, such as using `Time` in module imports without defining what `Time` is. The models also wrote constraints that mixed incompatible language features, which Alloy does not accept.

We believe that the weak model capabilities using PAT and Alloy are primarily because Alloy and PAT are much less popular than TLA$^+$ in real-world systems. Consequently, LLMs are not extensively trained on these languages, resulting in poor generation quality. These results justify the use of TLA$^+$ as the specification language of choice for SYSMOBENCH.

### B.3 COMPARISON WITH RELATED WORK

PAT-Agent (Zuo et al., 2025) and Alloy-APR (Alhanahnah et al., 2025) also evaluate AI's ability to work with formal models using PAT and Alloy, reporting promising results on their benchmarks. However, their tasks and complexity differ fundamentally from SYSMOBENCH. Table 5 summarizes the task of each work.

Table 5: Summary of the tasks of each work.

| Work | Task |
|---|---|
| SYSMOBENCH | Generating formal models for real-world software systems from their source code. |
| PAT-Agent | Generating formal models from natural language descriptions. |
| Alloy-APR | Repairing an existing model with injected errors. |

Because these tasks are inherently different, it is difficult to compare their complexity directly. Instead, we compare the complexity of the generated formal models as shown in Table 6.

Table 6: Complexity comparison across benchmarks measured by lines of code of formal models.

| Benchmark | Smallest | Largest | Median | Task Type |
|---|---|---|---|---|
| SYSMOBENCH | 75 | 508 | 219 | Generation |
| PAT-Agent | 16 | 142 | 45 | Generation |
| Alloy-APR (ARepair) | 15 | 99 | 50 | Repair |
| Alloy-APR (Alloy4Fun) | 1 | 234 | 21 | Repair |

Compared with the generation task in PAT-Agent, most models we expect LLMs/agents to generate in SYSMOBENCH are larger than the largest models in the PAT-Agent paper. PAT-Agent's tasks are small samples such as river-crossing puzzles and restaurant workflows, not real-world software systems. Alloy-APR's tasks are similar, which come from ARepair and Alloy4Fun; neither of them uses real-world system artifacts.

To further validate our understanding, we reproduced the results of Alloy-APR and PAT-Agent using the same LLMs evaluated in SYSMOBENCH. For Alloy-APR, we used the official artifact on the ARepair benchmark. Table 7 shows the results.

Table 7: Reproduction of Alloy-APR results on ARepair benchmark with LLMs used in SYS-MOBENCH.

| Model | Correct Items | Success Rate |
|---|---|---|
| Claude-Sonnet-4 | 38 / 38 | 100.0% |
| GPT-5 | 30 / 38 | 78.9% |
| Gemini-2.5-Pro | 14 / 38 | 36.8% |
| DeepSeek-R1 | 5 / 38 | 13.2% |
| Best result in Alloy-APR | 28 / 38 | 73.7% |

Our reproduction results show that Claude-Sonnet-4 and GPT-5 outperform the best results reported in the Alloy-APR paper. This suggests that existing LLMs can solve these repair tasks effectively–the high scores in the paper are largely due to the fact that the task itself is relatively simple. In contrast, our results show that these LLMs still struggle to generate syntax-correct Alloy models from complex system code in SYSMOBENCH (see Table 4).

For PAT-Agent, we ran the NoPlanning workflow using the LLMs evaluated in SYSMOBENCH. This workflow is similar to our Basic Modeling Agent: it calls the LLM to generate a PAT model and then iteratively fixes errors. Table 8 shows the results.

Table 8: Reproduction of PAT-Agent results using NoPlanning workflow with LLMs from SYS-MOBENCH. CSR: Compilation Success Rate, FPR: Full Pass Rate, APR: Average Pass Rate.

| Model | CSR | FPR | APR |
|---|---|---|---|
| Claude-Sonnet-4 | 84.6% | 80.8% | 87.3% |
| GPT-5 | 84.6% | 69.2% | 76.4% |
| Gemini-2.5-Pro | 84.6% | 65.4% | 74.8% |
| DeepSeek-R1 | 57.7% | 50.0% | 54.9% |

The results are consistent with the original paper's findings. Similar to Alloy-APR, current LLMs can solve these relatively simple tasks to a reasonable extent (e.g., Claude-Sonnet-4 achieves 87.3% APR). However, their ability to generate formal models for real-world software systems is much weaker, as evidenced by our results (see Table 4).

These results suggest that existing benchmarks such as PAT-Agent and Alloy-APR mostly exercise simplified modeling tasks. In contrast, SYSMOBENCH targets formal models derived from real system code, where current LLMs often fail to produce even syntax-correct specifications (see Table 4).

## C    QUALITATIVE EVALUATION OF AI-GENERATED MODELS

Beyond the automated quantitative metrics, we performed qualitative evaluation to assess the practical utility of AI-generated models for human engineers. We evaluated AI-generated models in two aspects: (1) comparison with human-written ground-truth models from the community, and (2) their ability to reproduce known bugs in system code.

### C.1    COMPARISON WITH HUMAN-WRITTEN MODELS

To assess the quality of AI-generated models, two human experts evaluated models produced by two different agents: the basic modeling agent and the code translation agent. Each expert compared AI-generated TLA$^{+}$ models against human-written models for nine of the systems in SYSMOBENCH.

The experts identified ten main types of differences between AI-generated and human models (Tables 9 and 10 summarize their occurrence across systems and LLMs):

1. Unnecessary `EXTENDS/INSTANCE` statements
2. Topics present in the human model but missing in AI models
3. Topics introduced by AI but absent in the human model
4. Fewer comments compared to human models
5. Properties present in human models but not in AI models
6. Different fairness assumptions compared to human models
7. Longer composite actions in AI models
8. Overly complex or random fairness conditions
9. Overspecialization with hard-coded values instead of parameters

Table 9: Types of differences between AI-generated models (produced by the basic modeling agent) and human-written models. Numbers refer to the types listed above.

| System | Claude-Sonnet-4 | GPT-5 | Gemini-2.5-Pro | DeepSeek-R1 |
|---|---|---|---|---|
| Asterinas Spin | 1, 3 | 1, 5, 8 | 1, 5, 8 | 1, 5, 8 |
| Asterinas Mutex | 1, 5 | 1, 5 | 1, 5, 9 | 5 |
| Asterinas Rwmutex | 1, 5 | 1, 5 | 1, 5 | 1, 5 |
| Etcd Raft | 1, 2, 4, 5, 7 | 1, 2, 4, 5 | 1, 2, 4, 5, 7 | 1, 2, 4, 5, 7 |
| Redis Raft | 1, 4 | 1, 4, 7 | 1, 4, 7 | 1, 4 |
| Xline CURP | 1, 2, 4, 5, 6 | 1, 2, 3, 4, 5, 6 | 1, 2, 4, 5, 6 | 1, 2, 4, 5, 6 |
| PGo dqueue | 1, 5 | 1, 5 | 1, 5 | 1, 5, 7 |
| PGo locksvc | 1, 5 | 1, 5 | 1, 5 | 5, 6, 7, 8 |
| PGo raftkvs | 1, 5, 7 | 1, 5, 7, 8 | 1, 5, 8 | 1, 5 |

Table 10: Types of differences between AI-generated models (produced by the code translation agent) and human-written models. Numbers refer to the types listed above.

| System | Claude-Sonnet-4 | GPT-5 | Gemini-2.5-Pro | DeepSeek-R1 |
|---|---|---|---|---|
| Asterinas Spin | 1, 2, 3, 5, 6 | 1, 3, 5, 6, 8 | 1, 2, 3, 5, 6 | 1, 2, 3, 5, 6, 8 |
| Asterinas Mutex | 1, 2, 5, 8 | 1, 2, 5 | 1, 2, 5 | 1, 2, 5, 8 |
| Asterinas Rwmutex | 1, 5, 6 | 1, 3, 5, 6, 8 | 1, 3, 5, 6 | 1, 5, 6, 8 |
| Etcd Raft | 1, 2, 4, 5, 6, 8 | 1, 2, 4, 5, 6 | 1, 2, 4, 5, 6, 7 | 1, 2, 4, 5, 6, 7 |
| Redis Raft | 1, 3, 4, 5 | 1, 3, 4, 5 | 1, 3, 4, 5 | 1, 2, 4, 5, 6 |
| Xline CURP | 1, 4, 5, 6 | 4, 5, 6 | 1, 2, 4, 5, 6 | 1, 4, 5, 6 |
| PGo dqueue | 1, 2, 3, 5, 6 | 1, 5, 6 | 1, 5, 6 | 1, 3, 5, 6 |
| PGo locksvc | 1, 5 | 1, 5, 8 | 1, 5 | 1, 5, 6, 8 |
| PGo raftkvs | 1, 5, 7 | 1, 5, 7, 8 | 1, 5, 6, 7, 8 | 1, 5, 6, 7 |

We group and discuss these differences below.

**Prompt-induced patterns (types 1, 4, 5).** For both agents, many AI models include unnecessary `EXTENDS / INSTANCE` statements (type 1), lack comments (type 4), and omit certain properties (type 5). These patterns largely result from our prompting and evaluation design. The prompt requires including common libraries to avoid syntax errors; this does not harm correctness or the evaluation of AI's modeling capability, as human experts also sometimes copy-paste `EXTENDS` with unnecessary dependencies. Missing comments and properties are expected, as SYSMOBENCH focuses on state/action modeling and does not require comment or property generation.

**Model utility (types 2, 3, 6, 8, 9).** AI-generated models may miss certain variables or actions (type 2) or include extra details (type 3), especially when there is a significant difference in abstraction levels between the human-written models and AI-generated models. For instance, the code translation agent tends to produce more concrete specifications compared to human-written ones, leading to more frequent occurrences of type 2 (missing topics) and type 3 (extra topics). Fairness definitions (types 6 and 8) of AI-generated models often differ from human models or are overly technical or random, which can affect liveness checking. There are also isolated cases of overspecialization (type 9). Overall, these differences show that AI models capture the general structure but may vary in completeness, fairness, and abstraction compared to human models.

**Readability and documentation (types 4, 7).** For both agents, the issue of fewer comments (type 4) is due to our prompt design; when we removed the instruction not to generate comments, AI models produced reasonably long comments that are easy to read. Some models also contain long composite actions (type 7) or use unconventional ordering of structure (e.g., TypeOK checks placed unusually). Nevertheless, AI-generated models from both agents remain generally readable, with meaningful variable and action names and understandable structure.

## C.2 BUG REPRODUCTION

AI-generated models can be practically useful for partial correctness checking. Without any hints about specific bugs, AI-generated models successfully reproduced several hard-to-find bugs across multiple systems in SYSMOBENCH. Table 11 lists these reproduced bugs with links to the corresponding issue reports or pull requests.

Table 11: Bugs successfully reproduced by AI-generated models.

| Bug ID | Description |
|---|---|
| Etcd Raft #10998 | Learners cannot vote during promotion causing election failure |
| Xline CURP #402 | Cluster will enter a frozen state after multiple crashes and recoveries |
| Redis Raft #19 | Stale reads under process pauses, violating linearizability |
| Asterinas Mutex #1279 | Failed try_lock incorrectly unlocks mutex breaking mutual exclusion |
| Asterinas Rwmutex #1303 | Lost wakeup when upgradeable reader releases lock |

These bugs were discovered by having AI agents generate TLA+ models from earlier versions of the system code and then using model checking to identify the issues.

## D EXAMPLES OF AI-GENERATED TLA+ MODELS

We present two AI-generated system models. Figures 5 and 6 show the TLA+ model and its corresponding TLC configuration for Asterinas Spinlock generated by the basic modeling agent with Claude-Sonnet-4. Figures 7 and 8 show the TLA+ model and its corresponding TLC configuration for Etcd Raft generated by the code translation agent with Claude-Sonnet-4.

```
1  ---- MODULE spin ----
2
3  EXTENDS TLC, Sequences, SequencesExt, Naturals, FiniteSets, Bags
4
5  CONSTANTS Threads
6
7  VARIABLES
8      lock_state,
9      thread_state,
10     guards
11
12 Vars == <<lock_state, thread_state, guards>>
13
14 TypeOK ==
15     /\ lock_state \in BOOLEAN
16     /\ thread_state \in [Threads -> {"idle", "spinning", "locked"}]
17     /\ guards \subseteq Threads
18
19 Init ==
20     /\ lock_state = FALSE
21     /\ thread_state = [t \in Threads |-> "idle"]
22     /\ guards = {}
23
24 TryAcquire(t) ==
25     /\ thread_state[t] \in {"idle", "spinning"}
26     /\ lock_state = FALSE
27     /\ lock_state' = TRUE
28     /\ thread_state' = [thread_state EXCEPT ![t] = "locked"]
29     /\ guards' = guards \cup {t}
30
31 StartSpin(t) ==
32     /\ thread_state[t] = "idle"
33     /\ lock_state = TRUE
34     /\ thread_state' = [thread_state EXCEPT ![t] = "spinning"]
35     /\ UNCHANGED <<lock_state, guards>>
```

```
36
37 SpinLoop(t) ==
38     /\ thread_state[t] = "spinning"
39     /\ lock_state = TRUE
40     /\ UNCHANGED <<lock_state, thread_state, guards>>
41
42 SpinAcquire(t) ==
43     /\ thread_state[t] = "spinning"
44     /\ lock_state = FALSE
45     /\ lock_state' = TRUE
46     /\ thread_state' = [thread_state EXCEPT ![t] = "locked"]
47     /\ guards' = guards \cup {t}
48
49 TryLock(t) ==
50     /\ thread_state[t] = "idle"
51     /\ IF lock_state = FALSE
52        THEN /\ lock_state' = TRUE
53             /\ thread_state' = [thread_state EXCEPT ![t] = "locked"]
54             /\ guards' = guards \cup {t}
55        ELSE /\ UNCHANGED <<lock_state, thread_state, guards>>
56
57 Unlock(t) ==
58     /\ thread_state[t] = "locked"
59     /\ t \in guards
60     /\ lock_state' = FALSE
61     /\ thread_state' = [thread_state EXCEPT ![t] = "idle"]
62     /\ guards' = guards \ {t}
63
64 Next ==
65     \E t \in Threads:
66         \/ TryAcquire(t)
67         \/ StartSpin(t)
68         \/ SpinLoop(t)
69         \/ SpinAcquire(t)
70         \/ TryLock(t)
71         \/ Unlock(t)
72
73 Spec == Init /\ [][Next]_Vars /\ \A t \in Threads: WF_Vars(Unlock(t))
74
75 ====
```

Figure 5: Asterinas Spinlock TLA$^+$ model generated by the basic modeling agent with Claude-Sonnet-4.

```
1 SPECIFICATION Spec
2
3 CONSTANTS
4     Threads = {t1, t2, t3}
```

Figure 6: Asterinas Spinlock TLC configuration generated by the basic modeling agent with Claude-Sonnet-4.

```
1  ---- MODULE etcdraft ----
2  EXTENDS TLC, Sequences, SequencesExt, Naturals, FiniteSets, Bags
3
4  CONSTANTS
5      Server,          \* Set of server IDs
6      Value,           \* Set of possible values for log entries
7      Nil,             \* Constant representing no value/no leader
8      Follower,        \* Node state constants
9      PreCandidate,
10     Candidate,
11     Leader
12
13 VARIABLES
14     state,           \* state[i] is the state of server i
15     currentTerm,     \* currentTerm[i] is the current term of server i
16     votedFor,        \* votedFor[i] is who server i voted for in currentTerm[i]
17     log,             \* log[i] is the log of server i
18     commitIndex,     \* commitIndex[i] is the commit index of server i
19     nextIndex,       \* nextIndex[i][j] is the next log index to send to server j from leader i
20     matchIndex,      \*matchIndex[i][j]isthehighestlogindexknowntobereplicatedonserverjbyleaderi
21     messages,        \* Set of messages in transit
22     electionTimeout, \* electionTimeout[i] tracks election timeout for server i
23     leader           \* leader[i] is the current leader known to server i
24
25 vars == <<state, currentTerm, votedFor, log, commitIndex, nextIndex, matchIndex, messages,
   ↪ electionTimeout, leader>>
```

```
26
27  \* Message types
28  MsgHup == "MsgHup"
29  MsgVote == "MsgVote"
30  MsgVoteResp == "MsgVoteResp"
31  MsgPreVote == "MsgPreVote"
32  MsgPreVoteResp == "MsgPreVoteResp"
33  MsgApp == "MsgApp"
34  MsgAppResp == "MsgAppResp"
35  MsgHeartbeat == "MsgHeartbeat"
36  MsgProp == "MsgProp"
37
38  \* Helper functions
39  Min(a, b) == IF a < b THEN a ELSE b
40  Max(a, b) == IF a > b THEN a ELSE b
41
42  LastTerm(xlog) == IF Len(xlog) = 0 THEN 0 ELSE xlog[Len(xlog)].term
43
44  Send(m) == messages' = messages \cup {m}
45
46  \* Initial state
47  Init ==
48      /\ state = [i \in Server |-> Follower]
49      /\ currentTerm = [i \in Server |-> 0]
50      /\ votedFor = [i \in Server |-> Nil]
51      /\ log = [i \in Server |-> <<>>]
52      /\ commitIndex = [i \in Server |-> 0]
53      /\ nextIndex = [i \in Server |-> [j \in Server |-> 1]]
54      /\ matchIndex = [i \in Server |-> [j \in Server |-> 0]]
55      /\ messages = {}
56      /\ electionTimeout = [i \in Server |-> 0]
57      /\ leader = [i \in Server |-> Nil]
58
59  \* Election timeout - triggers election
60  Timeout(i) ==
61      /\ state[i] \in {Follower, PreCandidate, Candidate}
62      /\ electionTimeout' = [electionTimeout EXCEPT ![i] = 0]
63      /\ state' = [state EXCEPT ![i] = IF state[i] = Follower THEN PreCandidate ELSE @]
64      /\ IF state[i] = Follower
65         THEN Send([type |-> MsgHup, from |-> i, to |-> i, term |-> currentTerm[i]])
66         ELSE messages' = messages
67      /\ UNCHANGED <<currentTerm, votedFor, log, commitIndex, nextIndex, matchIndex, leader>>
68
69  \* Start prevote campaign
70  StartPreVote(i) ==
71      /\ state[i] = PreCandidate
72      /\ state' = [state EXCEPT ![i] = PreCandidate]
73      /\ LET lastIndex == Len(log[i])
74             lastTerm == LastTerm(log[i])
75         IN messages' = messages \cup
76             {[type |-> MsgPreVote,
77               from |-> i,
78               to |-> j,
79               term |-> currentTerm[i] + 1,
80               lastLogIndex |-> lastIndex,
81               lastLogTerm |-> lastTerm] : j \in Server \ {i}}
82      /\ UNCHANGED <<currentTerm, votedFor, log, commitIndex, nextIndex, matchIndex, electionTimeout,
83         ↪  leader>>
84
85  \* Handle prevote response
85  HandlePreVoteResponse(i, m) ==
86      /\ m.type = MsgPreVoteResp
87      /\ m.to = i
88      /\ state[i] = PreCandidate
89      /\ m.term = currentTerm[i] + 1
90      /\ LET votes == {msg \in messages : msg.type = MsgPreVoteResp /\
91                                          msg.to = i /\
92                                          msg.term = currentTerm[i] + 1 /\
93                                          msg.voteGranted = TRUE}
94             voteCount == Cardinality({msg.from : msg \in votes}) + 1  \* +1 for self
95         IN IF voteCount > Cardinality(Server) \div 2
96            THEN /\ state' = [state EXCEPT ![i] = Candidate]
97                 /\ currentTerm' = [currentTerm EXCEPT ![i] = currentTerm[i] + 1]
98                 /\ votedFor' = [votedFor EXCEPT ![i] = i]
99                 /\ LET lastIndex == Len(log[i])
100                        lastTerm == LastTerm(log[i])
101                    IN messages' = (messages \ {m}) \cup
102                        {[type |-> MsgVote,
103                          from |-> i,
104                          to |-> j,
105                          term |-> currentTerm[i] + 1,
106                          lastLogIndex |-> lastIndex,
107                          lastLogTerm |-> lastTerm] : j \in Server \ {i}}
```

```
108                     /\ UNCHANGED <<log, commitIndex, nextIndex, matchIndex, electionTimeout, leader>>
109            ELSE /\ messages' = messages \ {m}
110                     /\ UNCHANGED <<state, currentTerm, votedFor, log, commitIndex, nextIndex, matchIndex,
                    ↪  electionTimeout, leader>>
111
112  \* Handle vote request
113  HandleVoteRequest(i, m) ==
114      /\ m.type \in {MsgVote, MsgPreVote}
115      /\ m.to = i
116      /\ LET logOk == \/ m.lastLogTerm > LastTerm(log[i])
117                      \/ /\ m.lastLogTerm = LastTerm(log[i])
118                         /\ m.lastLogIndex >= Len(log[i])
119             grant == /\ m.term >= currentTerm[i]
120                      /\ logOk
121                      /\ IF m.type = MsgVote
122                         THEN \/ votedFor[i] = Nil
123                              \/ votedFor[i] = m.from
124                         ELSE TRUE
125         IN /\ IF m.type = MsgVote /\ m.term > currentTerm[i]
126              THEN /\ state' = [state EXCEPT ![i] = Follower]
127                   /\ currentTerm' = [currentTerm EXCEPT ![i] = m.term]
128                   /\ votedFor' = [votedFor EXCEPT ![i] = IF grant THEN m.from ELSE Nil]
129                   /\ leader' = [leader EXCEPT ![i] = Nil]
130              ELSE IF m.type = MsgVote /\ grant
131                   THEN /\ votedFor' = [votedFor EXCEPT ![i] = m.from]
132                        /\ UNCHANGED <<state, currentTerm, leader>>
133                   ELSE UNCHANGED <<state, currentTerm, votedFor, leader>>
134            /\ Send([type |-> IF m.type = MsgVote THEN MsgVoteResp ELSE MsgPreVoteResp,
135                     from |-> i,
136                     to |-> m.from,
137                     term |-> IF m.type = MsgPreVote THEN m.term ELSE currentTerm'[i],
138                     voteGranted |-> grant])
139            /\ messages' = messages \ {m}
140            /\ UNCHANGED <<log, commitIndex, nextIndex, matchIndex, electionTimeout>>
141
142  \* Handle vote response
143  HandleVoteResponse(i, m) ==
144      /\ m.type = MsgVoteResp
145      /\ m.to = i
146      /\ state[i] = Candidate
147      /\ m.term = currentTerm[i]
148      /\ LET votes == {msg \in messages : msg.type = MsgVoteResp /\
149                                           msg.to = i /\
150                                           msg.term = currentTerm[i] /\
151                                           msg.voteGranted = TRUE}
152             voteCount == Cardinality({msg.from : msg \in votes}) + 1   \* +1 for self vote
153         IN IF voteCount > Cardinality(Server) \div 2
154            THEN /\ state' = [state EXCEPT ![i] = Leader]
155                 /\ leader' = [leader EXCEPT ![i] = i]
156                 /\ nextIndex' = [nextIndex EXCEPT ![i] = [j \in Server |-> Len(log[i]) + 1]]
157                 /\ matchIndex' = [matchIndex EXCEPT ![i] = [j \in Server |-> 0]]
158                 /\ LET entry == [term |-> currentTerm[i], value |-> Nil]
159                    IN /\ log' = [log EXCEPT ![i] = Append(@, entry)]
160                       /\ messages' = (messages \ {m}) \cup
161                             {[type |-> MsgApp,
162                               from |-> i,
163                               to |-> j,
164                               term |-> currentTerm[i],
165                               prevLogIndex |-> Len(log[i]),
166                               prevLogTerm |-> LastTerm(log[i]),
167                               entries |-> <<entry>>,
168                               leaderCommit |-> commitIndex[i]] : j \in Server \ {i}}
169                 /\ UNCHANGED <<currentTerm, votedFor, commitIndex, electionTimeout>>
170            ELSE /\ messages' = messages \ {m}
171                 /\ UNCHANGED <<state, currentTerm, votedFor, log, commitIndex, nextIndex, matchIndex,
                    ↪  electionTimeout, leader>>
172
173  \* Client request (leader only)
174  ClientRequest(i, v) ==
175      /\ state[i] = Leader
176      /\ LET entry == [term |-> currentTerm[i], value |-> v]
177             newLog == Append(log[i], entry)
178         IN /\ log' = [log EXCEPT ![i] = newLog]
179            /\ messages' = messages \cup
180                {[type |-> MsgApp,
181                  from |-> i,
182                  to |-> j,
183                  term |-> currentTerm[i],
184                  prevLogIndex |-> Len(log[i]),
185                  prevLogTerm |-> LastTerm(log[i]),
186                  entries |-> <<entry>>,
187                  leaderCommit |-> commitIndex[i]] : j \in Server \ {i}}
```

```
188        /\ UNCHANGED <<state, currentTerm, votedFor, commitIndex, nextIndex, matchIndex, electionTimeout,
    ↪  leader>>
189
190  \* Handle append entries
191  HandleAppendEntries(i, m) ==
192      /\ m.type = MsgApp
193      /\ m.to = i
194      /\ IF m.term > currentTerm[i]
195         THEN /\ currentTerm' = [currentTerm EXCEPT ![i] = m.term]
196              /\ state' = [state EXCEPT ![i] = Follower]
197              /\ votedFor' = [votedFor EXCEPT ![i] = Nil]
198              /\ leader' = [leader EXCEPT ![i] = m.from]
199         ELSE /\ UNCHANGED <<currentTerm, votedFor>>
200              /\ IF m.term = currentTerm[i]
201                 THEN /\ state' = [state EXCEPT ![i] = Follower]
202                      /\ leader' = [leader EXCEPT ![i] = m.from]
203                 ELSE UNCHANGED <<state, leader>>
204      /\ electionTimeout' = [electionTimeout EXCEPT ![i] = 0]
205      /\ LET logOk == \/ m.prevLogIndex = 0
206                      \/ /\ m.prevLogIndex <= Len(log[i])
207                         /\ log[i][m.prevLogIndex].term = m.prevLogTerm
208         IN IF logOk
209            THEN /\ log' = [log EXCEPT ![i] = SubSeq(@, 1, m.prevLogIndex) \o m.entries]
210                 /\ commitIndex' = [commitIndex EXCEPT ![i] = IF m.leaderCommit > commitIndex[i]
211                                                              THEN Min(m.leaderCommit, Len(log'[i]))
212                                                              ELSE commitIndex[i]]
213                 /\ Send([type |-> MsgAppResp,
214                          from |-> i,
215                          to |-> m.from,
216                          term |-> currentTerm'[i],
217                          success |-> TRUE,
218                          matchIndex |-> m.prevLogIndex + Len(m.entries)])
219            ELSE /\ Send([type |-> MsgAppResp,
220                          from |-> i,
221                          to |-> m.from,
222                          term |-> currentTerm'[i],
223                          success |-> FALSE,
224                          matchIndex |-> 0])
225                 /\ UNCHANGED <<log, commitIndex>>
226      /\ messages' = messages \ {m}
227      /\ UNCHANGED <<nextIndex, matchIndex>>
228
229  \* Handle append response
230  HandleAppendResponse(i, m) ==
231      /\ m.type = MsgAppResp
232      /\ m.to = i
233      /\ state[i] = Leader
234      /\ m.term = currentTerm[i]
235      /\ IF m.success
236         THEN /\ matchIndex' = [matchIndex EXCEPT ![i][m.from] = m.matchIndex]
237              /\ nextIndex' = [nextIndex EXCEPT ![i][m.from] = m.matchIndex + 1]
238              /\ LET Agree(idx) == {i} \cup {s \in Server : matchIndex'[i][s] >= idx}
239                     agreeIndexes == {idx \in 1..Len(log[i]) :
240                                         Cardinality(Agree(idx)) > Cardinality(Server) \div 2 /\
241                                         log[i][idx].term = currentTerm[i]}
242                 IN commitIndex' = [commitIndex EXCEPT ![i] = IF agreeIndexes /= {}
243                                                              THEN Max(commitIndex[i], CHOOSE idx \in
        ↪  agreeIndexes :
244                                                                       \A idx2 \in agreeIndexes : idx >=
        ↪  idx2)
245                                                              ELSE commitIndex[i]]
246         ELSE /\ nextIndex' = [nextIndex EXCEPT ![i][m.from] = Max(1, nextIndex[i][m.from] - 1)]
247              /\ UNCHANGED <<matchIndex, commitIndex>>
248      /\ messages' = messages \ {m}
249      /\ UNCHANGED <<state, currentTerm, votedFor, log, electionTimeout, leader>>
250
251  \* Send heartbeat
252  SendHeartbeat(i) ==
253      /\ state[i] = Leader
254      /\ messages' = messages \cup
255          {[type |-> MsgHeartbeat,
256            from |-> i,
257            to |-> j,
258            term |-> currentTerm[i],
259            leaderCommit |-> commitIndex[i]] : j \in Server \ {i}}
260      /\ UNCHANGED <<state, currentTerm, votedFor, log, commitIndex, nextIndex, matchIndex,
    ↪  electionTimeout, leader>>
261
262  \* Handle heartbeat
263  HandleHeartbeat(i, m) ==
264      /\ m.type = MsgHeartbeat
265      /\ m.to = i
266      /\ IF m.term >= currentTerm[i]
```

```
267        THEN /\ currentTerm' = [currentTerm EXCEPT ![i] = m.term]
268             /\ state' = [state EXCEPT ![i] = Follower]
269             /\ votedFor' = [votedFor EXCEPT ![i] = Nil]
270             /\ leader' = [leader EXCEPT ![i] = m.from]
271             /\ electionTimeout' = [electionTimeout EXCEPT ![i] = 0]
272             /\ commitIndex' = [commitIndex EXCEPT ![i] = IF m.leaderCommit > commitIndex[i]
273                                                          THEN Min(m.leaderCommit, Len(log[i]))
274                                                          ELSE commitIndex[i]]
275        ELSE UNCHANGED <<currentTerm, state, votedFor, leader, electionTimeout, commitIndex>>
276     /\ messages' = messages \ {m}
277     /\ UNCHANGED <<log, nextIndex, matchIndex>>
278
279 \* Advance election timeout
280 AdvanceElectionTimeout(i) ==
281     /\ state[i] \in {Follower, PreCandidate, Candidate}
282     /\ electionTimeout' = [electionTimeout EXCEPT ![i] = @ + 1]
283     /\ UNCHANGED <<state, currentTerm, votedFor, log, commitIndex, nextIndex, matchIndex, messages,
    ↪  leader>>
284
285 Next ==
286     \/ \E i \in Server : Timeout(i)
287     \/ \E i \in Server : StartPreVote(i)
288     \/ \E i \in Server, m \in messages : HandlePreVoteResponse(i, m)
289     \/ \E i \in Server, m \in messages : HandleVoteRequest(i, m)
290     \/ \E i \in Server, m \in messages : HandleVoteResponse(i, m)
291     \/ \E i \in Server, v \in Value : ClientRequest(i, v)
292     \/ \E i \in Server, m \in messages : HandleAppendEntries(i, m)
293     \/ \E i \in Server, m \in messages : HandleAppendResponse(i, m)
294     \/ \E i \in Server : SendHeartbeat(i)
295     \/ \E i \in Server, m \in messages : HandleHeartbeat(i, m)
296     \/ \E i \in Server : AdvanceElectionTimeout(i)
297
298 Fairness ==
299     /\ \A i \in Server : WF_vars(Timeout(i))
300     /\ \A i \in Server : WF_vars(StartPreVote(i))
301     /\ \A i \in Server : WF_vars(\E m \in messages : HandlePreVoteResponse(i, m))
302     /\ \A i \in Server : WF_vars(\E m \in messages : HandleVoteRequest(i, m))
303     /\ \A i \in Server : WF_vars(\E m \in messages : HandleVoteResponse(i, m))
304     /\ \A i \in Server : WF_vars(\E m \in messages : HandleAppendEntries(i, m))
305     /\ \A i \in Server : WF_vars(\E m \in messages : HandleAppendResponse(i, m))
306     /\ \A i \in Server : WF_vars(SendHeartbeat(i))
307     /\ \A i \in Server : WF_vars(\E m \in messages : HandleHeartbeat(i, m))
308     /\ \A i \in Server : WF_vars(AdvanceElectionTimeout(i))
309
310 Spec == Init /\ [][Next]_vars /\ Fairness
311
312 ====
```

Figure 7: Etcd Raft TLA$^+$ model generated by the code translation agent with Claude-Sonnet-4.

```
1  SPECIFICATION Spec
2
3  CONSTANTS
4      Server = {s1, s2, s3}
5      Value = {v1, v2}
6      Nil = "Nil"
7      Follower = "Follower"
8      PreCandidate = "PreCandidate"
9      Candidate = "Candidate"
10     Leader = "Leader"
```

Figure 8: Etcd Raft TLC configuration generated by the code translation agent with Claude-Sonnet-4.

# E    PGO-COMPILED SYSTEMS

Table 1 lists all the system artifacts in SYSMOBENCH. Unlike other open-source systems implemented mostly by human developers, PGo systems represent a special kind of compiler-generated systems. PGo is a compiler converting distributed systems specifications written in a DSL of TLA$^+$ into executable systems implementations in Go (Hackett et al., 2023a).

These systems reflect production use cases:

- dqueue is a simple distributed queue with producers and consumers, which represents a common cloud computing mechanism. Similar distributed queues are available from many cloud platforms, like Amazon SQS, Cloudflare Queues, or Apache Kafka.

- locksvc is a simple distributed locking system, which represents a common distributed systems concept.

- raftkvs is a verified distributed key-value store, with competitive performance. For its consensus implementation, raftkvs specifies Raft (Ongaro & Ousterhout, 2014).

These systems are complex, each requiring several person-days of effort to specify. The raftkvs store is particularly complex, requiring almost a person-month of effort. While they are developed using a formal modeling language, these systems also account for practical coding concerns. Each system compiles to usable, non-trivial Go code. Notably, raftkvs outperforms other formally verified key-value stores, with 41% higher throughput than the next-best formally verified store implementation, and similar latency but 21% of the throughput achieved by Etcd.

A challenge unique to system modeling is that PGo-compiled systems contain machine-generated Go code, which includes unusual abstractions and coding patterns. For instance, the generated code makes extensive use of abstractions from PGo's runtime support library, while containing many synthetically named variables. These issues are representative of realistic engineering scenarios, such as generated code (macros, parser generators, state machines), or situations where the original source code is lost (decompilation artifacts). This type of source code input currently leads to poor performance on our benchmarks.

**PGo Trace Validation.** For AI-generated system models, we must validate their behavior against gathered execution traces. PGo's TraceLink feature provides a different trace validation method than for hand-written systems, allowing for automatic implementation tracing and TLA$^+$ glue generation. As a result, no additional work is needed to gather traces. For simplicity, we use traces taken from TraceLink's published artifact. From these traces, TraceLink is able to generate its own binding TLA$^+$, mapping these logs precisely to a TLA$^+$ state space.

# F    SYSMOBENCH EVALUATION PROMPTS

During SYSMOBENCH evaluation, LLMs are invoked for conformance and invariant correctness evaluation to extract information, map actions and variables, and concretize invariants based on invariant templates.

We show the complete prompts used for benchmark evaluation. These prompts are templates with parameterized fields that are instantiated by task-specific information. For demonstration, we instantiate the fields in the task for modeling Etcd Raft, with the instantiated parts marked in green.

## F.1    CONFORMANCE EVALUATION PROMPTS

Two prompts extract model information and map model action and variable names to code, both generating configuration files for script processing to support trace validation.

**Model component extraction.** This prompt directs an LLM to extract TLA$^+$ model components, such as constants, variables, and actions, which are used by a script to generate a *trace specification* (Cirstea et al., 2024). A trace specification constrains state space exploration along the code trace path to verify whether a model state space path matches the code trace. In the prompt, the {source_code} field is instantiated with the TLA$^+$ model.

---

**Model Component Extraction Prompt**

```
Generate a YAML configuration file from the provided TLA+ model (.tla) and configuration (.cfg) files.
Extract information according to the following rules:

## Task Description
Parse the TLA+ model and configuration files to create a structured YAML configuration that captures the
model name, constants, variables, actions, and interactions.

## Extraction Rules
```

---

```
### spec_name
Extract from the module declaration line:  `---- MODULE <ModuleName> ----`
The spec_name is the ModuleName between "---- MODULE" and "----".

### constants
Extract from the CONSTANTS section in the .cfg file.
- name:  The constant identifier
- value:  The assigned value, formatted as:
- Sets:  Wrap in single quotes, e.g., '{s1, s2, s3}' becomes '{"s1", "s2", "s3"}'
- Strings:  Wrap in single quotes with double quotes inside, e.g., Nil becomes '"Nil"'
- Numbers:  Wrap in single quotes as string, e.g., 5 becomes '5'

### variables
Extract from the Init operator definition in the .tla file.
For each variable assignment in Init:
- name:  The variable name
- default_value:  The initial value expression (preserve TLA+ syntax, escape backslashes)

### actions
Extract from the Next operator definition.  Include only direct action calls (not numbered interactions).
For each action:
- name:  The action/operator name
- parameters:  List of parameters with:
- name:  Parameter variable name
- source:  Where the parameter comes from (e.g., Server, messages)
- stmt:  The complete statement as it appears in Next (including any conditions)

### interactions
Extract from the Next operator definition.  Include only numbered intermediate actions.
Just list the names (e.g., HandletickElection_1, HandletickHeartbeat_1)

## Example
Given this TLA+ model:

---- MODULE SimpleSpec ----
...
Init ==
/\ x = 0
/\ y = [s \in Server |-> 0]

Next ==
\/ \E s \in Server :  Action1(s)
\/ \E m \in messages :  Action2(m)
\/ IntermediateAction_1

And this configuration:

CONSTANTS
Server = {s1, s2}
MaxValue = 10

Generate this YAML:

```yaml
spec_name:  SimpleSpec
constants:
- name:  Server
value:  '{"s1", "s2"}'
- name:  MaxValue
value:  '10'
variables:
- name:  x
default_value:  '0'
- name:  y
default_value:  '[s \\in Server |-> 0]'
actions:
- name:  Action1
parameters:
- name:  s
source:  Server
stmt:  Action1(s)
- name:  Action2
parameters:
- name:  m
source:  messages
stmt:  Action2(m)
```

```
interactions:
- name: IntermediateAction_1
```

## Important Notes
1. Return ONLY the YAML content - no explanations, comments, or natural language
2. Preserve TLA+ syntax exactly in default_value fields (escape backslashes)
3. For actions with conditions, include the full stmt as it appears in Next
4. Ignore variables that appear in Init but are not part of the main model (e.g., pc, info, stack)
5. Order matters: spec_name, constants, variables, actions, interactions

Generate the YAML configuration based on the provided TLA+ files:

{source_code}
```

**Model and code component mapping.** This prompt directs an LLM to map code to model variable naming for trace validation comparison. The LLM generates a JSON file storing the mappings, which is further processed by a script to output a test harness that aligns code trace variable and action names with the model. In the prompt, the {TLA_SPEC_CODE_PLACEHOLDER} field is instantiated with the TLA$^+$ model, and {IMPLEMENTATION_CODE_PLACEHOLDER} with the corresponding code.

### Model and Code Component Mapping Prompt

```
You are tasked with generating a JSON mapping file that defines how to convert a concurrent or
distributed system traces to TLA+ model format for trace validation.

## System Overview

etcd Raft is a distributed consensus algorithm implementation that supports:
- Leader election with terms and prevoting/voting
- Log replication across multiple nodes
- State transitions between Follower, Candidate, and Leader roles
- Message passing between nodes

## Code Analysis
Before generating the mapping, you need to analyze the relevant code to understand the system behavior:

**CRITICAL**: You MUST base your mapping on the actual TLA+ model content, NOT on the examples below.
The examples are for format reference only. Always use the actual variables and actions defined in the
provided model.

### TLA+ Model Code
```tla+
{TLA_SPEC_CODE_PLACEHOLDER}
```

### Implementation Code
```go
{IMPLEMENTATION_CODE_PLACEHOLDER}
```

## Input: System Trace Format

System traces are in JSONL format with events like:
```json
{"conf": [["1", "2", "3"], []], "log": 0, "name": "InitState", "nid": "1", "role": "StateFollower",
"state": {"commit": 0, "term": 0, "vote": "0"}}
{"conf": [["1", "2", "3"], []], "log": 1, "name": "BecomeCandidate", "nid": "1", "role":
"StateCandidate", "state": {"commit": 0, "term": 1, "vote": "0"}}
{"conf": [["1", "2", "3"], []], "log": 1, "name": "BecomeCandidate", "nid": "2", "role":
"StateCandidate", "state": {"commit": 0, "term": 1, "vote": "0"}}
```

Common actions in system traces:
- BecomeFollower: Transition to follower role
- BecomeCandidate: Transition to candidate role
- BecomeLeader: Transition to leader role
- Ready: Node is ready for operations
- PreVote/Vote: Cast prevote/vote during election
- AppendEntries: Replicate log entries
- Heartbeat: Send/receive heartbeat messages

## Target: TLA+ Model Variables
```

```
The TLA+ model tracks these state variables:
- currentTerm:  Current term number for each node
- state:  Node role (Follower, Candidate, Leader)
- votedFor:  Which candidate this node voted for in current term
- commitIndex:  Index of highest log entry known to be committed
- nextIndex:  For leaders, next log entry to send to each server
- matchIndex:  For leaders, highest log entry known to be replicated on server

## Required Mapping Structure

Generate a JSON file with this structure:

```json
{
"config":  {
"Server":  ["Server1", "Server2", "Server3"] // List of node identifiers
},
"events":  {
// Map system actions to TLA+ events
"InitState":  "Init",
"BecomeFollower":  "BecomeFollower",
"BecomeCandidate":  "BecomeCandidate",
"BecomeLeader":  "BecomeLeader",
"Ready":  "Ready",
"Vote":  "Vote",
"AppendEntries":  "AppendEntries",
"Heartbeat":  "Heartbeat",
// Add other mappings as needed based on code analysis
},
"node_mapping":  {
// Map string node IDs to node names
"1":  "Node1",
"2":  "Node2",
"3":  "Node3",
// Continue as needed
},
"role_mapping":  {
// Map system roles to TLA+ states
"StateFollower":  "Follower",
"StateCandidate":  "Candidate",
"StateLeader":  "Leader"
}
}
```

## Implementation Notes
1.  The mapping will be used by a state tracker that maintains complete system state
2.  Server IDs in traces are numeric (0, 1, 2...)  and must be mapped to "Server1", "Server2", etc.
3.  The state tracker will automatically handle state transitions based on actions
4.  Focus on correctly mapping actions and Server states
5.  The config section should list all possible Server that might appear in traces

## Your Task
Generate a complete mapping.json file that:
1.  Maps all common actions to their TLA+ equivalents
2.  Provides server ID mappings for all servers that appear in traces
3.  Ensures compatibility with the state tracking implementation
```

## F.2 INVARIANT CORRECTNESS EVALUATION PROMPT

This prompt concretizes invariants from given invariant templates to model-specific forms. It typically requires the LLM to map different names in an invariant template to the corresponding model elements. The $tla_model field is instantiated with the TLA$^+$ model, and the $invariant_templates field with the invariant templates defined in the system artifact (see §3.2.4 for an example).

---

**Invariant Concretization Prompt**

```
You are a TLA+ expert specializing in distributed systems and Raft consensus.  Your task is to implement
a set of expert-written invariants for the given etcd TLA+ model.

## Target Model
```

```
$tla_model

## Invariants to Implement

$invariant_templates

## Implementation Requirements

1.  **Deep Analysis**:  First, thoroughly understand both the invariant template's semantic intent
and the model's modeling approach:
- What distributed consensus property does each template aim to verify?
- How does the model represent server states, logs, terms, and leadership?
- What are the semantic equivalents between template concepts and model implementation?

2.  **Semantic Mapping**:  For each invariant, identify the conceptual mapping between template and
model:
- Template server state concepts -> Model's server state representation
- Template log structure -> Model's log data structures and indexing
- Template leadership concepts -> Model's leader election and term management
- Template node/server sets -> Model's server constants and domains

3.  **Creative Adaptation**:  Translate the invariant while preserving its core safety/liveness meaning:
- **DO NOT** simply replace variable names - understand the underlying distributed systems logic
- **DO** redesign the predicate logic to fit the model's data structure granularity
- **DO** use equivalent semantic concepts even if data representations differ
- **PRESERVE** the original safety/liveness guarantees without weakening the property

4.  **TLA+ Property Type Constraints**:

**FOR SAFETY PROPERTIES** (type:  "safety"):
- **MUST** be STATE PREDICATES (describe single states only)
- **NEVER** use primed variables (`currentTerm'`, `log'`)
- **NEVER** use temporal operators (`[]`, `<>`, `~>`)
- **NEVER** reference actions (like `RequestVote(s)`, `AppendEntries(s,t)`) - only use state variables
- **ONLY** use unprimed variables (`currentTerm[s]`, `log[s]`) and constants
- **CORRECT**:  `LeaderUniqueness == \A term \in Terms :  Cardinality({s \in Servers :  state[s] =
"leader" /\ currentTerm[s] = term}) <= 1`
- **INCORRECT**:  `state[s] = "candidate" => RequestVote(s)` (references action RequestVote)

**FOR LIVENESS PROPERTIES** (type:  "liveness"):
- **MUST** be TEMPORAL FORMULAS (describe execution traces)
- **MUST** use temporal operators (`<>`, `~>`) to express "eventually" or "leads-to"
- **CORRECT**:  `EventualLeaderElection == <>(\E s \in Servers :  state[s] = "leader")`

5.  **Constraint Compliance**:
- Use ONLY variables, constants, and operators that exist in the model
- Generate complete, syntactically valid TLA+ invariant definitions
- Maintain the exact invariant names from templates

6.  **Output format**:  Return a JSON object containing an array of complete TLA+ invariant definitions

7.  **EXACT naming requirement**:  You MUST use the exact invariant names specified in the templates
above.  Do not create your own names.

## Example Output Format

```json
{
"invariants":  [
"LeaderUniqueness == \\A term \\in 1..MaxTerm :  Cardinality({n \\in Servers :  state[n].role =
\"leader\" /\\ state[n].currentTerm = term}) <= 1",
"LogConsistency == \\A n1, n2 \\in Servers :  \\A i \\in DOMAIN log[n1] :  (i \\in DOMAIN log[n2] /\\
log[n1][i].term = log[n2][i].term) => (\\A j \\in 1..i :  log[n1][j] = log[n2][j])"
]
}
```

**CRITICAL REQUIREMENTS**:
- **SEMANTIC PRESERVATION**:  Each translated invariant MUST verify the same property as the original
template
- **CREATIVE ADAPTATION**:  Do NOT simply omit invariants - find creative ways to express the same
property using available model elements
- **COMPLETENESS**:  Aim to translate ALL invariants by understanding their semantic intent, not just
their syntactic form
- Use ONLY variables, constants, and operators that exist in the provided model
- Use EXACTLY the invariant names from the templates (preserve exact names for evaluation consistency)
```

```
- Return ONLY valid JSON, no explanatory text before or after
- Each array element must be a complete TLA+ invariant definition:  "InvariantName == <expression>"
- For complex invariants, you may use multiline format within the JSON string (use actual line breaks)
- For simple invariants, single line format is preferred
- **LAST RESORT**:  Only omit an invariant if its core concept is fundamentally incompatible with the
model's design
- **CRITICAL JSON ESCAPING RULES**:
- TLA+ operators like `\A`, `\E`, `\in` contain ONE backslash in the final TLA+ code
- In JSON strings, use EXACTLY ONE backslash escape:  write `"\\A"` to get `\A` in TLA+
- **DO NOT double-escape**:  `"\\\\A"` is WRONG and will produce `\\A` in TLA+
- **CORRECT**:  `"LeaderUniqueness == \\A term \\in 1..MaxTerm :  state[term] = \"leader\""`
- **WRONG**:  `"LeaderUniqueness == \\\\A term \\\\in 1..MaxTerm :  state[term] = \"leader\""`
- Start your response immediately with the opening brace {
```

## G  BASIC MODELING AGENT

The basic modeling agent operates in two steps for each system artifact: (1) generating the model, including both the TLA$^+$ model and its TLC configuration, and (2) using a feedback loop that takes SYSMOBENCH evaluation results to iteratively improve the generated TLA$^+$ model. We show the complete prompts of the basic modeling agent (§4) to provide its detailed implementation.

### G.1  MODEL GENERATION PROMPTS

**TLA$^+$ model generation.**    This prompt directs an LLM to generate the TLA$^+$ model file, instantiated with the granularity definitions of the system artifact (see §3.1). The {file_path} and {source_code} fields are instantiated with the code file path in the repository and source code content, respectively.

---
**TLA$^+$ Model Generation Prompt**

```
You are an expert in formal verification and TLA+ models with deep expertise in concurrent and
distributed systems, particularly etcd and Raft consensus
.

Convert the following source code to a comprehensive TLA+ model.

System:  etcd distributed key-value store

Source Code from {file_path}:
```go
{source_code}
```

System-specific modeling requirements:

MANDATORY CORE ACTIONS (must include all):
1.   [Message Types] MsgHup (election timeout), MsgVote/MsgVoteResp (voting), MsgApp/MsgAppResp (log
replication)
2.   [Node States] Four states:  StateFollower, StateCandidate, StateLeader, StatePreCandidate (prevote
enabled)
3.   [Leader Election] Complete prevote + vote phases:  PreCandidate → Candidate → Leader transitions
4.   [Log Operations] Log entry appending, consistency checks, commitment with majority quorum
5.   [Heartbeat/Timeout] Election timeouts triggering campaigns, heartbeat prevention of elections
6.   [Client Proposals] MsgProp message handling and log entry creation by leaders

EXPLICITLY EXCLUDED (do not model):
- Configuration changes and joint consensus (ConfChange messages)
- Log compaction and snapshots (MsgSnap)
- ReadIndex optimizations (MsgReadIndex)
- Async storage operations (LocalAppendThread, LocalApplyThread)
- Advanced flow control and progress tracking details

REQUIRED BEHAVIORAL SCOPE:
- Prevote phase (StatePreCandidate) must be modeled as it's enabled by default in etcd
- State transition constraints:  Follower → PreCandidate → Candidate → Leader (strict transitions)
- Message processing by state:  only valid message types handled in each node state
- Term advancement rules:  nodes advance term when receiving messages with higher term
- Voting restrictions:  one vote per term, term must be current or newer
- Heartbeat mechanism:  leaders send heartbeats, followers reset election timeout on receipt
- Log consistency checks:  prevLogIndex/prevLogTerm validation in MsgApp processing
```
---

```
- Majority-based leader election and log commitment
- Basic network message delays and losses

Generate a TLA+ model that accurately models the system's behavior.

CRITICAL OUTPUT REQUIREMENTS:
1.  The MODULE name must be exactly "etcdraft
" (---- MODULE etcdraft ----)

2.  Return ONLY pure TLA+ model code - no markdown code blocks (no ```tla or ```)
3.  Do not include any explanations, comments, or formatting markers
4.  Start your response directly with:  ---- MODULE etcdraft
----
5.  End your response with the closing ====
6.  **DO NOT define invariants** (like MutualExclusion, Invariant, etc.)  - focus on modeling the system
behavior
7.  **MUST include EXTENDS statement**:  The model must extend at least these modules:  TLC, Sequences,
SequencesExt, Naturals, FiniteSets, Bags
```

**TLC configuration generation.** This prompt directs an LLM to generate a TLC configuration file. The configuration file requires the LLM's understanding of the system to make the model executable, such as designating the initial predicate and next-state relations. The $tla_spec field is instantiated with the TLA$^+$ model generated in the previous step.

**TLC Configuration Generation Prompt**

```
You are a TLA+ expert.  Generate a complete TLC configuration file (.cfg) for the etcd model that can be
directly saved and used for model checking.

## Input Model:

$tla_spec

## Requirements:

1.  **Analyze the model** to identify the main model name and all declared constants
2.  **Generate complete .cfg file content** with SPECIFICATION, CONSTANTS sections
3.  **Use small values for constants** to ensure efficient model checking (2-3 servers, small integers)
4.  **Output ONLY the raw .cfg file content** - no explanations, no markdown, no code blocks

## Example Output Format:

SPECIFICATION SpecName

CONSTANTS
...

**CRITICAL: Your response must contain exactly ONE complete .cfg file.  Do not repeat any sections.
Start your response immediately with "SPECIFICATION" and include nothing else.**
```

## G.2   MODEL REFINEMENT PROMPT

This prompt provides guidance for the LLM to refine the previously generated model using syntax and runtime evaluation results from SYSMOBENCH. The {current_model} field contains the previous iteration's model, {current_tlc_cfg} contains the previous TLC configuration, {syntax_errors} contains the syntax errors reported by SANY, and {runtime_errors} contains the runtime errors reported by TLC.

**Model Refinement Prompt**

```
You are an expert TLA+ model specialist with extensive experience in concurrent and distributed systems
modeling.

I need you to fix errors in a TLA+ model for etcdraft system.

## Current TLA+ Model
```tla
{current_model}
```

## Current TLC Configuration
```

```
```
{current_tlc_cfg}
```

## Errors Found

**Detailed Syntax Errors:**
{syntax_errors}

**Detailed Runtime Errors:**
{runtime_errors}

## Correction Instructions
This is correction attempt {attempt_number} of {max_attempts}.

Please provide a corrected TLA+ model that fixes these errors.  Your corrected model should:

1.  **Fix all syntax errors**:  Ensure proper TLA+ syntax, correct operator usage, and valid module
structure
2.  **Resolve runtime errors**:  Define missing variables, operators, and ensure logical consistency
3.  **Maintain original intent**:  Keep the core distributed system logic and behavior from the source
code
4.  **Follow TLA+ best practices**:  Use appropriate data structures, actions, and invariants
5.  **Be complete and self-contained**:  Include all necessary EXTENDS, CONSTANTS, VARIABLES, and
operator definitions

Focus specifically on:
- Defining any missing variables or constants
- Implementing missing operators or functions
- Fixing syntax issues with operators, expressions, or module structure
- Ensuring proper action definitions and state transitions
- Maintaining consistency with etcdraft's system behavior

**CRITICAL OUTPUT REQUIREMENTS:**
- Return ONLY pure TLA+ model code
- NO markdown code blocks (no ```tla or ```)
- NO explanations, comments, or text outside the model
- NO formatting markers of any kind
- The MODULE name must be exactly "etcdraft"
- Start directly with:  ---- MODULE etcdraft ----
```

## H    TRACE LEARNING AGENT

The trace learning agent does not use any code as input; instead, it relies on the distributed traces as context.  Similar to the basic modeling agent, we provide an initial prompt analogous to the basic modeling agent's prompt (§G.1), but substituting the codebase context with trace information instead. If the first model generation fails to pass compilation, the model refinement loop will pass the errors back to the LLM to iteratively fix the model.

**Trace formats.** The trace-based method works with several types of traces and can be easily extended to additional systems. For each trace format, we provide a short custom prompt explaining the format. We currently support:

- `.ndjson` and `.jsonl` logs: Newline-delimited JSON, with coarse-grained logs defined by the specific system. One log file contains multiple nodes' execution logs.

- PGo-instrumented logs generated by TraceLink (Hackett & Beschastnikh, 2025): Also newline-delimited JSON and contains PGo-specific concepts like archetype names and vector clocks. Variable updates are logged in fine-grained detail at each PGo-defined critical section. One log file is output per node; there are multiple log files per distributed execution.

**Optimizations.** We anecdotally noticed that passing single execution traces results in overfitting by the model, with generated models closely reflecting the single executed path. Providing more traces improves context for the model.

One issue we encountered was fitting large traces into models' context windows. The JSON structure of traces is expensive in tokens, because each "[", ":", and other punctuation represents a separate token.  Most of the models we used had a context window of about 200K tokens; a JSON trace of several megabytes, such as the Etcd Raft traces, simply could not fit. We solved this with three workarounds:

- We support sampling for systems with large traces, randomly choosing a set of execution traces among all collected traces.

- We convert the nested JSON structure into tab-separated values (TSV) format, which deduplicates JSON keys into the TSV header and uses only tabs as a separator to save tokens.

- We abbreviate repeated state or action values (e.g., `ReceiveRequestVoteResponse`) to acronyms (e.g., RRVR) and provide a mapping in the prompt.

The TSV and abbreviation optimizations significantly save tokens: with the Claude tokenizer, it reduces token use by 62% for ten lines of Etcd Raft traces (from 645 to 262 tokens), and by 63% for ten lines of mutex traces (from 866 to 318 tokens). This enabled us to fit multiple traces into the initial prompt, reducing the impact of overfitting. We did not apply this optimization to the other methods, since code is less structured and does not have obvious candidates for deduplication.

## I  COMPLETE EVALUATION RESULTS

### I.1  LIVENESS VIOLATION ANALYSIS

We analyzed counterexamples from two representative systems (Asterinas SpinLock and Etcd Raft) and categorized liveness violations into two main classes:

- *Fairness-related issues* that prevent progress due to missing fairness declarations, overly narrow or overly broad constraints (e.g., defined as WF(Next));

- *Logical/structural issues* that block progress due to conflicting updates or missing/incorrect logic in action definitions.

Since the modeling task focuses on state/action models of the system implementation and LLMs are not required to generate temporal operators (e.g., in liveness properties), our categorization does not include errors related to temporal operators.

Table 12 presents the detailed breakdown of violations by category and LLM. For Asterinas Spin-Lock, fairness-related issues dominate the violations, particularly "too broad" and "too narrow" constraints. For instance, Claude-Sonnet-4, generated 26 out of 32 violations due to overly broad fairness assumptions.

For Etcd Raft, liveness violations are primarily caused by logical/structural issues. The model's large state space causes these logical errors to block progress before fairness-related issues can manifest. Nevertheless, manual inspection confirms that fairness conditions are generally incorrect.

Table 12: Liveness violations by category in Asterinas SpinLock and Etcd Raft for the basic modeling agent.

(a) Asterinas SpinLock liveness violations by category

| LLM | Fairness too broad | Fairness too narrow | Missing fairness | Logical errors | Total violations |
|---|---|---|---|---|---|
| Claude-Sonnet-4 | 26 | 2 | 4 | 0 | 32 |
| GPT-5 | 8 | 10 | 2 | 0 | 20 |
| Gemini-2.5-Pro | 4 | 6 | 0 | 0 | 10 |
| DeepSeek-R1 | 4 | 2 | 0 | 2 | 8 |

(b) Etcd Raft liveness violations by category

| LLM | Logical missing/errors | Conflicting updates | Total violations |
|---|---|---|---|
| Claude-Sonnet-4 | 4 | 8 | 12 |
| GPT-5 | 2 | 8 | 20 |
| Gemini-2.5-Pro | 0 | 0 | 0 |
| DeepSeek-R1 | 4 | 4 | 8 |

## I.2 DETAILED RESULTS BY SYSTEM

We present the complete evaluation results for all systems in our benchmark using three AI agents: Basic Modeling, Code Translation, and Trace Learning in Tables 13–23. These tables follow the same evaluation setup as described in §4.

Table 13: Asterinas Spinlock

| Agent | LLM | Syntax | Runtime | Conformance | Invariant |
|---|---|---|---|---|---|
| Basic Modeling | Claude-Sonnet-4 | 100.00% ✓ | 100.00% ✓ | 100.00% | 100.00% |
| | GPT-5 | 100.00% ✓ | 100.00% ✓ | 80.00% | 100.00% |
| | Gemini-2.5-Pro | 100.00% ✓ | 100.00% ✓ | 80.00% | 85.71% |
| | DeepSeek-R1 | 100.00% ✓ | 100.00% ✓ | 80.00% | 100.00% |
| Code Translation | Claude-Sonnet-4 | 100.00% ✓ | 100.00% ✓ | 100.00% | 100.00% |
| | GPT-5 | 100.00% ✓ | 100.00% ✓ | 100.00% | 85.71% |
| | Gemini-2.5-Pro | 100.00% ✓ | 100.00% ✓ | 100.00% | 100.00% |
| | DeepSeek-R1 | 100.00% ✓ | 100.00% ✓ | 100.00% | 100.00% |
| Trace Learning | Claude-Sonnet-4 | 50.00% ✗ | - | - | - |
| | GPT-5 | 100.00% ✓ | 0.00% ✗ | - | - |
| | Gemini-2.5-Pro | 100.00% ✓ | 0.00% ✗ | - | - |
| | DeepSeek-R1 | 100.00% ✓ | 0.00% ✗ | - | - |

The trace learning agent underperforms compared to the other two agents, typically failing compilation and runtime checks. We observe that it is more difficult for LLMs to process structured trace data, in comparison to source code. Specifically, Claude-Sonnet-4 appears to be particularly weak in this regard, achieving the lowest syntax scores, despite its coding capabilities. This trend of the trace learning agent is consistent across all the evaluated system artifacts (Tables 14–23).

Table 14: Etcd Raft

| Agent | LLM | Syntax | Runtime | Conformance | Invariant |
|---|---|---|---|---|---|
| Basic Modeling | Claude-Sonnet-4 | 100.00% ✓ | 25.00% ✓ | 7.69% | 69.23% |
| | GPT-5 | 47.87% ✗ | - | - | - |
| | Gemini-2.5-Pro | 50.00% ✗ | - | - | - |
| | DeepSeek-R1 | 50.00% ✗ | - | - | - |
| Code Translation | Claude-Sonnet-4 | 100.00% ✓ | 66.67% ✓ | 15.38% | 92.31% |
| | GPT-5 | 100.00% ✓ | 20.00% ✗ | - | - |
| | Gemini-2.5-Pro | 44.44% ✗ | - | - | - |
| | DeepSeek-R1 | 100.00% ✓ | 0.00% ✗ | - | - |
| Trace Learning | Claude-Sonnet-4 | 50.00% ✗ | - | - | - |
| | GPT-5 | 48.78% ✗ | - | - | - |
| | Gemini-2.5-Pro | 42.31% ✗ | - | - | - |
| | DeepSeek-R1 | 47.73% ✗ | - | - | - |

Table 15: Asterinas Mutex

| Agent | LLM | Syntax | Runtime | Conformance | Invariant |
|---|---|---|---|---|---|
| Basic Modeling | Claude-Sonnet-4 | 100.00% ✓ | 100.00% ✓ | 100.00% | 100.00% |
| | GPT-5 | 100.00% ✓ | 100.00% ✓ | 100.00% | 85.71% |
| | Gemini-2.5-Pro | 100.00% ✓ | 100.00% ✓ | 66.67% | 85.71% |
| | DeepSeek-R1 | 100.00% ✓ | 100.00% ✓ | 66.67% | 100.00% |
| Code Translation | Claude-Sonnet-4 | 100.00% ✓ | 100.00% ✓ | 100.00% | 100.00% |
| | GPT-5 | 100.00% ✓ | 100.00% ✓ | 100.00% | 100.00% |
| | Gemini-2.5-Pro | 100.00% ✓ | 100.00% ✓ | 100.00% | 85.71% |
| | DeepSeek-R1 | 100.00% ✓ | 100.00% ✓ | 100.00% | 85.71% |
| Trace Learning | Claude-Sonnet-4 | 50.00% ✗ | - | - | - |
| | GPT-5 | 50.00% ✗ | - | - | - |
| | Gemini-2.5-Pro | 100.00% ✓ | 0.00% ✗ | - | - |
| | DeepSeek-R1 | 50.00% ✗ | 0.00% ✗ | - | - |

Table 16: Asterinas Rwmutex

| Agent | LLM | Syntax | Runtime | Conformance | Invariant |
|---|---|---|---|---|---|
| Basic Modeling | Claude-Sonnet-4 | 100.00% ✓ | 100.00% ✓ | 100.00% | 90.00% |
| | GPT-5 | 100.00% ✓ | 100.00% ✓ | 75.00% | 80.00% |
| | Gemini-2.5-Pro | 100.00% ✓ | 100.00% ✓ | 0.00% | 80.00% |
| | DeepSeek-R1 | 100.00% ✓ | 100.00% ✓ | 50.00% | 90.00% |
| Code Translation | Claude-Sonnet-4 | 100.00% ✓ | 100.00% ✓ | 100.00% | 90.00% |
| | GPT-5 | 100.00% ✓ | 100.00% ✓ | 100.00% | 90.00% |
| | Gemini-2.5-Pro | 100.00% ✓ | 100.00% ✓ | 100.00% | 80.00% |
| | DeepSeek-R1 | 100.00% ✓ | 100.00% ✓ | 50.00% | 90.00% |
| Trace Learning | Claude-Sonnet-4 | 50.00% ✗ | - | - | - |
| | GPT-5 | 100.00% ✓ | 0.00% ✗ | - | - |
| | Gemini-2.5-Pro | 100.00% ✓ | 0.00% ✗ | - | - |
| | DeepSeek-R1 | 50.00% ✗ | - | - | - |

Table 17: Asterinas Ringbuffer

| Agent | LLM | Syntax | Runtime | Conformance | Invariant |
|---|---|---|---|---|---|
| Basic Modeling | Claude-Sonnet-4 | 100.00% ✓ | 100.00% ✓ | 100.00% | 100.00% |
| | GPT-5 | 100.00% ✓ | 0.00% ✗ | - | - |
| | Gemini-2.5-Pro | 100.00% ✓ | 0.00% ✗ | - | - |
| | DeepSeek-R1 | 100.00% ✓ | 0.00% ✗ | - | - |
| Code Translation | Claude-Sonnet-4 | 100.00% ✓ | 100.00% ✓ | 100.00% | 100.00% |
| | GPT-5 | 100.00% ✓ | 100.00% ✓ | 100.00% | 75.00% |
| | Gemini-2.5-Pro | 100.00% ✓ | 100.00% ✓ | 100.00% | 100.00% |
| | DeepSeek-R1 | 100.00% ✓ | 0.00% ✗ | - | - |
| Trace Learning | Claude-Sonnet-4 | 100.00% ✓ | 0.00% ✗ | - | - |
| | GPT-5 | 100.00% ✓ | 0.00% ✗ | - | - |
| | Gemini-2.5-Pro | 100.00% ✓ | 0.00% ✗ | - | - |
| | DeepSeek-R1 | 100.00% ✓ | 0.00% ✗ | - | - |

Table 18: Redis Raft

| Agent | LLM | Syntax | Runtime | Conformance | Invariant |
|---|---|---|---|---|---|
| Basic Modeling | Claude-Sonnet-4 | 100.00% ✓ | 0.00% ✗ | - | - |
| | GPT-5 | 100.00% ✓ | 0.00% ✗ | - | - |
| | Gemini-2.5-Pro | 50.00% ✗ | - | - | - |
| | DeepSeek-R1 | 100.00% ✓ | 0.00% ✗ | - | - |
| Code Translation | Claude-Sonnet-4 | 100.00% ✓ | 23.81% ✓ | 9.09% | 75.00% |
| | GPT-5 | 100.00% ✓ | 0.00% ✗ | - | - |
| | Gemini-2.5-Pro | 50.00% ✗ | - | - | - |
| | DeepSeek-R1 | 100.00% ✓ | 100.00% ✓ | 0.00% | 25.00% |
| Trace Learning | Claude-Sonnet-4 | 50.00% ✗ | - | - | - |
| | GPT-5 | 47.06% ✗ | - | - | - |
| | Gemini-2.5-Pro | 100.00% ✓ | 0.00% ✗ | - | - |
| | DeepSeek-R1 | 48.53% ✗ | - | - | - |

The system model generated by DeepSeek-R1 is overly simplified; thus, it has high syntax and runtime correctness, but has low score on invariants (the protocol logic is incorrect) and 0% on conformance.

Table 19: Xline CURP

| Agent | LLM | Syntax | Runtime | Conformance | Invariant |
|---|---|---|---|---|---|
| Basic Modeling | Claude-Sonnet-4 | 100.00% ✓ | 0.00% ✗ | - | - |
| | GPT-5 | 100.00% ✓ | 0.00% ✗ | - | - |
| | Gemini-2.5-Pro | 100.00% ✓ | 0.00% ✗ | - | - |
| | DeepSeek-R1 | 100.00% ✓ | 0.00% ✗ | - | - |
| Code Translation | Claude-Sonnet-4 | 100.00% ✓ | 100.00% ✓ | 50.00% | 100.00% |
| | GPT-5 | 100.00% ✓ | 0.00% ✗ | - | - |
| | Gemini-2.5-Pro | 100.00% ✓ | 100.00% ✓ | 66.67% | 100.00% |
| | DeepSeek-R1 | 100.00% ✓ | 0.00% ✗ | - | - |
| Trace Learning | Claude-Sonnet-4 | 50.00% ✗ | - | - | - |
| | GPT-5 | 100.00% ✓ | 0.00% ✗ | - | - |
| | Gemini-2.5-Pro | 46.15% ✗ | - | - | - |
| | DeepSeek-R1 | 100.00% ✓ | 0.00% ✗ | - | - |

Xline CURP is one of the largest system artifacts in SYSMOBENCH (see Table 1). We suspect that the system model generated by Gemini-2.5-Pro benefits from its 1M-token context window, enabling effective summarization of the 4000+ line codebase into a concise TLA$^+$ representation.

Table 20: PGo dqueue

| Agent | LLM | Syntax | Runtime | Conformance | Invariant |
|---|---|---|---|---|---|
| Basic Modeling | Claude-Sonnet-4 | 100.00% ✓ | 33.33% ✓ | 33.33% | 0.00% |
| | GPT-5 | 100.00% ✓ | 0.00% ✗ | - | - |
| | Gemini-2.5-Pro | 100.00% ✓ | 0.00% ✗ | - | - |
| | DeepSeek-R1 | 100.00% ✓ | 0.00% ✗ | - | - |
| Code Translation | Claude-Sonnet-4 | 100.00% ✓ | 100.00% ✓ | 0.00% | 100.00% |
| | GPT-5 | 100.00% ✓ | 100.00% ✓ | 0.00% | 100.00% |
| | Gemini-2.5-Pro | 100.00% ✓ | 100.00% ✓ | 0.00% | 100.00% |
| | DeepSeek-R1 | 100.00% ✓ | 100.00% ✓ | 0.00% | 100.00% |
| Trace Learning | Claude-Sonnet-4 | 100.00% ✓ | 0.00% ✗ | - | - |
| | GPT-5 | 100.00% ✓ | 0.00% ✗ | - | - |
| | Gemini-2.5-Pro | 100.00% ✓ | 0.00% ✗ | - | - |
| | DeepSeek-R1 | 100.00% ✓ | 0.00% ✗ | - | - |

Table 21: PGo locksvc

| Agent | LLM | Syntax | Runtime | Conformance | Invariant |
|---|---|---|---|---|---|
| Basic Modeling | Claude-Sonnet-4 | 44.45% ✗ | - | - | - |
| | GPT-5 | 100.00% ✓ | 0.00% ✗ | - | - |
| | Gemini-2.5-Pro | 100.00% ✓ | 0.00% ✗ | - | - |
| | DeepSeek-R1 | 100.00% ✓ | 0.00% ✗ | - | - |
| Code Translation | Claude-Sonnet-4 | 100.00% ✓ | 100.00% ✓ | 0.00% | 83.33% |
| | GPT-5 | 100.00% ✓ | 100.00% ✓ | 0.00% | 66.67% |
| | Gemini-2.5-Pro | 100.00% ✓ | 0.00% ✗ | - | - |
| | DeepSeek-R1 | 100.00% ✓ | 100.00% ✓ | 0.00% | 50.00% |
| Trace Learning | Claude-Sonnet-4 | 42.31% ✗ | - | - | - |
| | GPT-5 | 100.00% ✓ | 0.00% ✗ | - | - |
| | Gemini-2.5-Pro | 50.00% ✗ | - | - | - |
| | DeepSeek-R1 | 50.00% ✗ | - | - | - |

Table 22: PGo raftkvs

| Agent | LLM | Syntax | Runtime | Conformance | Invariant |
|-------|-----|--------|---------|-------------|-----------|
| Basic Modeling | Claude-Sonnet-4 | 44.57% ✗ | - | - | - |
| | GPT-5 | 100.00% ✓ | 0.00% ✗ | - | - |
| | Gemini-2.5-Pro | 45.84% ✗ | - | - | - |
| | DeepSeek-R1 | 45.32% ✗ | - | - | - |
| Code Translation | Claude-Sonnet-4 | 100.00% ✓ | 50.00% ✓ | 0.00% | 90.91% |
| | GPT-5 | 100.00% ✓ | 100.00% ✓ | 0.00% | 72.73% |
| | Gemini-2.5-Pro | 40.91% ✗ | - | - | - |
| | DeepSeek-R1 | 100.00% ✓ | 22.22% ✗ | - | - |
| Trace Learning | Claude-Sonnet-4 | 50.00% ✗ | - | - | - |
| | GPT-5 | 46.55% ✗ | - | - | - |
| | Gemini-2.5-Pro | 41.67% ✗ | - | - | - |
| | DeepSeek-R1 | 47.83% ✗ | - | - | - |

We observe that the characteristics of LLM performance on PGo-compiled systems are very different from human-written systems as discussed in Section 5 and Appendix E. We find that GPT-5 generally performs well on PGo systems, indicating its ability of understanding machine-generated code patterns.

Table 23: ZooKeeper Fast Leader Election (FLE)

| Agent | LLM | Syntax | Runtime | Conformance | Invariant |
|-------|-----|--------|---------|-------------|-----------|
| Basic Modeling | Claude-Sonnet-4 | 100.00% ✓ | 0.00% ✗ | - | - |
| | GPT-5 | 100.00% ✓ | 0.00% ✗ | - | - |
| | Gemini-2.5-Pro | 100.00% ✓ | 0.00% ✗ | - | - |
| | DeepSeek-R1 | 100.00% ✓ | 0.00% ✗ | - | - |
| Code Translation | Claude-Sonnet-4 | 100.00% ✓ | 0.00% ✗ | - | - |
| | GPT-5 | 100.00% ✓ | 0.00% ✗ | - | - |
| | Gemini-2.5-Pro | 100.00% ✓ | 0.00% ✗ | - | - |
| | DeepSeek-R1 | 100.00% ✓ | 0.00% ✗ | - | - |
| Trace Learning | Claude-Sonnet-4 | 44.44% ✗ | - | - | - |
| | GPT-5 | 47.92% ✗ | - | - | - |
| | Gemini-2.5-Pro | 100.00% ✓ | 0.00% ✗ | - | - |
| | DeepSeek-R1 | 100.00% ✓ | 0.00% ✗ | - | - |

ZooKeeper FLE has the largest codebase and implements the complex ZAB protocol, making it the most challenging system to model among all eleven artifacts.

