# OpenReview forum: "SysMoBench: Evaluating AI on Formally Specifying Complex Real-World Systems"
_ICLR.cc/2026/Conference — ICLR 2026 Poster_

### Official Review · Reviewer_RwMC · 2025-10-19

**Soundness:** 3
**Presentation:** 3
**Contribution:** 2
**Rating:** 2
**Confidence:** 4

**Summary:**

In this work, the authors propose a benchmark dataset for evaluating LLMs in terms of their capability of system modelling. The dataset contains 9 systems/models, and the authors propose 4 different metrics for evaluating the quality of the model constructed by the LLMs. Some empirical results are then presented.

**Strengths:**

On the positive side, I enjoy reading the draft for the following reasons.

First, the problem of evaluating LLMs capability of system modeling, which can be regarded as one particular form of abstract thinking capability, is an interesting and important one.

Second, the idea of evaluating the quality of the models at different levels is a valid idea.

Lastly, the writing is easy to follow and some of the empirical results are interesting.

**Weaknesses:**

On the less positive side, the draft can be improved from the following aspects.

First, the benchmark data is fairly small in size and lacks variety (i.e., only one modeling language), and there is not a very interesting or useful benchmark yet. More importantly, it seems to be rather hard to scale up the dataset as well.

Second, the proposed ways of measuring the quality of the model can be much improved, and much details on how exactly the scoring is done is missing. For instance, for per-action specification, it is not clear how a specification is considered correct - do you have to check specification equivalence, modular renaming, using different data structures and so on?

The following are a list of detailed comments.

Page 2: “SYSMOBENCH currently includes a diverse set of nine real-world artifacts, …”

Comment: The size of the benchmark set raises questions on whether such a benchmark set would produce statistically significant evaluating results.

Page 2: “SYSMOBENCH enables us to understand the capabilities and limitations of AI in modeling realworld
systems by evaluating different agent designs with various AI models.”

Comment: Given that only one modelling language is evaluated, this statement is rather an over-statement.

Section 3.1: line 183-190

Comment: It is rather arguable whether such a modeling requirement is reasonable or. In practice, the level of details to be included in the model typically depends on the analysis task. i.e., the properties to be established or falsified. Simply saying that some actions should be modeled is not sufficient as there are many ways of modeling the same action.

Page 5: “It then uses SANY to check the per-action specification.”

Comment: How is this done? In general, this boils down to a specification equivalence checking which is hard.

Page 9: “For example, adding Etcd Raft took one SYSMOBENCH author four days; an Xline CURP contributor with no experience of SYSMOBENCH added the system to SYSMOBENCH in four days.”

Comment: Four days per mode is not exactly efficient. Furthermore, there are questions such as who decides the model is correct? Lastly, if four man/day is all that is needed for one model, why are there only 9 in total?

**Questions:**

If the AI uses different data-structures (one-dimensional list instead of two-dimensional one) or different encoding to model the same action,  can you tell that it is correct?

---

> ### Author Response · Authors · 2025-11-21
> **We have addressed your comments in the new version.**
>
> Thanks for the insightful review. We appreciate it.
>
> We have addressed the comments and uploaded the new version. Please check it out.
>
> In the rebuttal, we first answer the question and then respond to the weakness comments.
>
> ---
>
> **> Q1: If the AI uses different data-structures (one-dimensional list instead of two-dimensional one) or different encoding to model the same action, can you tell that it is correct?**
>
> The answer is yes. As long as the formal model faithfully describes the system, i.e., the model conforms to the system code, then SysMoBench considers the model to be correct, no matter what data structures the model uses.
>
> There is a misunderstanding based on the following comments
>
> > Page 5: “It uses SANY to check the per-action specification.”
>
> > Comment: How is this done? In general, this boils down to a specification equivalence checking which is hard.
>
> SANY is used for syntax checking as a metric of syntax correctness. It does not perform equivalence checking.
>
> ---
>
> **> C1: The benchmark is fairly small in size**
>
> We agree that nine systems do not look big. In fact, this reflects the key challenges of developing real-world system-level AI benchmarks beyond code-snippet-level benchmarks – despite many coding benchmarks, there are few system benchmarks, because system artifacts are large, complex, and hard to prepare, especially by benchmark developers who were not the developers of the systems. As a result, system intelligence of existing LLMs is weak.
>
> We believe that the only way to make progress is to gradually develop real-world system-level benchmarks through a community-based approach, where the system developers add their own system artifacts into the benchmark. SysMoBench shows such feasibility. In fact, the system artifacts we added in the benchmark were those we ourselves have been working on for years. That’s the reason that it only takes us 3-4 days to add one artifact. For systems that we never worked on or are unfamiliar with, it will take much longer.
>
> Note that this is the same process as many popular benchmarks were developed. For example, the original SWE-bench only uses 12 open-source project repositories and it has been growing ever since.
>
> ---
>
> **> C2: The benchmark data lacks variety -- only one modeling language**
>
> The reason we chose TLA+ as the one language is from a practical standpoint. TLA+ is the primary and _de facto_ formal specification language for distributed and concurrent systems in practice, is widely used by software companies like Amazon, Microsoft, Nvidia, Google, Oracle, etc to check and verify critical infrastructure systems such as distributed consensus systems like Etcd and ZooKeeper, confidential consortium frameworks, databases, OS kernel synchronization, and cache coherence. It is not unfair to say that there is no better alternative than TLA+ with respect to formally modeling real-world computing systems.
>
> As suggested by Reviewer K1Um, we extended SysMoBench to support two other specification languages, Alloy and PAT. We used the basic modeling agent to generate the system model in PAT and Alloy. The AI-generated Alloy and PAT models are rather poor – only GPT-5 can generate an Alloy model that passes the syntax correctness check, after many attempts. We believe that the weak model capabilities using PAT and Alloy are mostly because Alloy and PAT are much less popular than TLA+ in real-world systems; so LLMs are not extensively trained on them. This also justify the use of TLA+.
>
> ---
>
> **> C3: Details on how exactly the scoring is done is missing**
>
> Thank you for the feedback. We have added the details in the new version.
>
> We clarify the per-action specification check is only for syntax check. In general, SysMoBench focuses on behavior conformance rather than structural equivalence.
>
> ---
>
> **> C4: It is rather arguable whether such a modeling requirement is reasonable**
>
> This is true – the level of details depends on the analysis tasks. So, for the same system artifact, we can create different problems by asking for different levels of details.
>
> The task we’re focusing on is model checking based formal verification. We require the same level of details as in prior work on verification and bug finding of these systems.
>
> Our principles are:
> * The model can check real code and find real bugs. Hence, it must include actions that interact with other components. We define mandatory core actions and exclude implementation‑level details, as shown in line 183-190.
> * We evaluate behavioral conformance rather than structural equivalence. The required level of detail is defined _apriori_ based on what is minimally necessary for a system model to support implementation‑level reasoning such as conformance and invariant. If an AI‑generated model omits these core behaviors, it lacks the semantic obligations needed to represent the system effectively, thus failing to enable further system verification.
> * We allow finer‑grained modeling of mandatory actions.

---

> > ### Comment · Reviewer_RwMC · 2025-11-26
> >
> > Thanks for the response and for admitting some of the limitations.
> >
> > Regarding "As long as the formal model faithfully describes the system", from what I can tell, it would require a level of formal modeling which would be unnecessarily detailed - for instance, to provide a model insisting on using one-dimensional list (instead of two-dimensional), and would be problematic.

---

> > > ### Author Response · Authors · 2025-11-26
> > > **Clarify the misunderstanding**
> > >
> > > Dear Reviewer RwMC,
> > >
> > > There appear to be a misunderstanding of typical TLA+ model usage – **SysMoBench does not require anything different from how TLA+ for system modeling is carried out in practice.**
> > >
> > > Basically, SysMoBench does not require the generated formal models to be structurally equivalent to the real system code; hence, SysMoBench does not require unnecessarily detailed models.
> > >
> > > The mentioned case – one-dimensional vs two-dimensional data structure – could both pass the conformance checking, because **SysMoBench does not require the use of any particular data structures.**
> > >
> > > For example, in the `locksvc` system, GPT-5 generates a one-slot mailbox model (a 1-dimensional array), while Claude Sonnet 4 generates a FIFO-queue model (a 2-dimensional array). Both types of models can pass our evaluation.
> > >
> > > The conformance checking process in SysMoBench is not any sort of structural evaluation. Instead, **conformance evaluates behavioral conformance.** That is, we consider whether the formal model we ask AI to generate includes appropriately ordered actions whose effects respect the specified invariants.
> > >
> > > Basically, in conformance checking, we compare two things:
> > >
> > > **1. Action sequences**
> > >
> > > The traces generated from the real system only include the minimal actions required for the system’s basic functionality.
> > >
> > > For example, for a spinlock, traces contain only TryLock, Lock, and Unlock operations.
> > >
> > > During comparison, we use the Missing-Event technique (see Tang et al., 2025) which executes additional spec transitions between the observed actions. This design ensures that a coarse spec (modeling only TryLock, Lock and Unlock), and a fine-grained spec (with extra steps such as internal Wait transitions) will both be accepted as long as the TryLock/Lock/Unlock ordering is preserved. This guarantees that a different modeling granularity does not affect correctness.
> > >
> > > **2. Core-variable state comparison**
> > >
> > > We classify variables into core and non-core.
> > >
> > > Core variables are essential to the protocol, e.g., Raft's state and term or mutex/spinlock's lock flag. These variables are intentionally simple and semantically clear. For comparison, we use LLMs to translate the system trace state values into the spec’s representation. This means regardless of how the spec encodes these simple variables, we can still compare them.
> > >
> > > Non-core variables, such as message buffers or log entries, are not compared, so their modeling styles can vary widely across specifications and do not affect correctness.
> > >
> > > Moreover, formal models are evaluated with respect to certain invariants. As long as different formal models satisfy the same abstraction that an invariant expresses, both models will pass model checking.
> > >
> > > **The above principles have been used consistently across research on using TLA+ formal models for model checking of system code**, such as (Cirstea et al., 2024, Hackett & Beschastnikh, 2025; Howard et al., 2025; Ouyang et al., 2025; Tang et al., 2024; Tang et al., 2025). Traditionally, these formal models were hand-written (which often takes months to years). In this work, we use LLMs/agents to generate these formal models by relying on the same underlying principles used in these prior manual efforts.
> > >
> > > **Please let us know if the above answer addresses your concern or anything that is unclear to you.**
> > >
> > > Thanks!

---

### Official Review · Reviewer_K1Um · 2025-10-19

**Soundness:** 2
**Presentation:** 3
**Contribution:** 2
**Rating:** 6
**Confidence:** 4

**Summary:**

The paper introduces SYSMOBENCH, a benchmark designed to evaluate generative AI (LLMs and agents) on their ability to formally model complex real-world systems, particularly concurrent and distributed ones.
Using TLA+ as the modeling language, the benchmark assesses how well AI can synthesize system-level formal specifications from code, documentation, and execution traces.

SYSMOBENCH defines four automatic evaluation metrics and provides tasks from nine real-world system artifacts (e.g., Etcd Raft, Redis Raft, Asterinas OS spinlock/mutex).
Experiments with four LLMs (Claude-Sonnet-4, GPT-5, Gemini-2.5-Pro, DeepSeek-R1) and three agent designs show that AI can handle simple concurrency models (e.g., spinlocks) but fails on large-scale distributed systems (e.g., Raft).

**Strengths:**

- Novel Benchmark and Scope

Focusing on system-level formal modeling, going beyond prior works on function-level specification or toy TLA+ examples.

- Automatic and Quantitative Evaluation Metrics

Proposes a clear, reproducible metric suite (syntax, runtime, conformance, invariant) for assessing AI-generated TLA+ models without human involvement.

- Empirical and Realistic Evaluation

Uses real-world system artifacts (Etcd, Redis, Asterinas, etc.), showing the practical limitations of current LLMs and agents in understanding and abstracting complex system behaviors.

**Weaknesses:**

- Shallow Semantic Understanding

The current metrics emphasize syntactic, runtime, and invariant correctness, but do not directly measure whether AI truly captures the semantic intent or design rationale of the system.

- Lack of Human Baseline Comparison

No comparison with expert-written TLA+ specifications is provided. Without a human reference, it is hard to assess how “good” the AI-generated models truly are in abstraction quality or maintainability.

- Limited Scalability

Adding a new system using LLM still requires 3–4 days of human setup (instrumentation, trace harness preparation), which challenges scalability despite claims of automation.

- Risk of Contamination and Overfitting

Some system code (e.g., Etcd Raft) may already appear in LLM training data, possibly affecting fairness; the paper lacks a clear anti-contamination discussion.

**Questions:**

How easy is it to extend SYSMOBENCH to other formal languages (e.g., Alloy, PAT)? A discussion on generalization beyond TLA+ would make the benchmark more widely applicable.

The authors note that LLMs often fail on liveness invariants. Could they provide finer-grained analysis (e.g., which temporal operators or fairness conditions are most misunderstood)?

Can SYSMOBENCH-produced AI-generated models be useful to human engineers (e.g., partial correctness checking or documentation)? Some qualitative human evaluation would strengthen its practical impact.

---

> ### Author Response · Authors · 2025-11-21
> **We have addressed your questions in the new version.**
>
> Thanks for the insightful review. We appreciate it.
>
> We have addressed the comments and uploaded the new version. Please check it out.
>
> In the rebuttal, we first answer the question and then respond to the weakness comments.
>
> We will use multiple comments as there are a lot of new content. Thanks for reading it.
>
> ---
>
> **> Q1: How easy is it to extend SYSMOBENCH to other formal languages (e.g., Alloy, PAT)?**
>
> Thanks for raising the question of extensibility. The SysMoBench framework is indeed made general to other specification languages like Alloy and PAT. We originally did not pursue this mainly because the ability of existing LLMs to use Alloy and PAT is rather poor, so it does not yield very meaningful results when the LLMs/agents cannot generate reasonable-quality models. Therefore, we focused on TLA+ in the benchmark.
>
> To prove the extensibility, during the rebuttal period, we added the support for Alloy and PAT. The implementation details can be found in Appendix B.1 and B.2.
>
> We used the basic modeling agent to generate the system model in PAT and Alloy. The AI-generated Alloy and PAT models are rather poor – only GPT-5 can generate an Alloy model that passes the syntax correctness check, after many attempts. We believe that the weak model capabilities using PAT and Alloy are mostly because Alloy and PAT are much less popular than TLA+ in real-world systems; so LLMs are not extensively trained on them. The detailed discussion is at Appendix B.3.
>
> ---
>
> **> Q2: Could they provide finer-grained analysis on liveness**
>
> Yes! We present the fine-grained analysis in Appendix I.1.
>
> We analyzed counterexamples from two representative systems (Asterinas SpinLock and Etcd Raft) and categorized violations into two main classes:
> * Fairness‑related issues (prevent progress): missing declaration, too narrow or too broad (defined as WF(Next)).
> * Logical/structural issues (block progress): conflicting updates, or logical missing/errors.
>
> Note that the modeling task focuses on state/action models of the system implementation; LLMs are not required to generate temporal operators (e.g., in liveness properties). Therefore, "temporal-operator misuse" is not applicable.
>
> Asterinas SpinLock liveness violations:
> |                 | Fairness too broad | Fairness too narrow | Missing fairness declaration | Logical errors | Total violations |
> |-----------------|:--------------------:|:--------------------:|:--------------------:|:--------------------:|:--------------------:|
> | Claude-Sonnet-4 | 26                 | 2                   | 4                            | 0              | 32               |
> | GPT-5           | 8                  | 10                  | 2                            | 0              | 20               |
> | Gemini-2.5-Pro  | 4                  | 6                   | 0                            | 0              | 10               |
> | DeepSeek-R1     | 4                  | 2                   | 0                            | 2              | 8                |
>
> In SpinLock, fairness-related issues dominate, particularly "too broad" and "too narrow", reflecting LLM's limitations in temporal reasoning.
>
> Etcd Raft liveness violations:
> |                 | Logical missing/errors | Conflicting updates | Total violations |
> |-----------------|:----------------------:|:-------------------:|:----------------:|
> | Claude-Sonnet-4 |            4           |          8          |        12        |
> | GPT-5           |            2           |          8          |        20        |
> | Gemini-2.5-Pro  |            0           |          0          |         0        |
> | DeepSeek-R1     |            4           |          4          |         8        |
>
> In Etcd Raft, liveness violations are caused by logical/structural issues rather than fairness assumptions. This is because the model's large state space causes these logical errors to block progress before fairness issues manifest. Nevertheless, manual inspection confirms that fairness conditions are generally incorrect.

---

> ### Author Response · Authors · 2025-11-21
> **We have addressed your questions in the new version (Cont.)**
>
> **> Q3: Can SYSMOBENCH-produced AI-generated models be useful to human engineers (e.g., partial correctness checking or documentation)? Some qualitative human evaluation would strengthen its practical impact.**
>
> Thanks for raising this point. We have added the detailed analysis in Appendix C of the updated version. We present some high-level results below.
>
> We performed qualitative human evaluation in two aspects:
> * qualitative comparison with human-written ground-truth models,
> * evaluating AI-generated models’ ability to reproduce known bugs in system code. While AI-generated models differ from human models in structure, content, and readability, they remain meaningful, partially correct, and useful for human engineers. The bug reproduction experiment provides direct evidence of practical utility.
>
> **1. Qualitative comparison with human-written models**
>
> Two human experts compared each auto-generated TLA+ model (produced by the Basic Modeling Agent and the Code Translation Agent, the same formal models evaluated in the paper) against the human-written models for all nine systems. The expert summarized nine main types of differences between AI-generated models and human-written models:
>
> 1. Unnecessary EXTENDS / INSTANCE statements
> 2. Topics present in the human model but missing in AI models
> 3. Topics introduced by AI but absent in the human model
> 4. Less comments compared to human models
> 5. Properties present in human models but not in AI models
> 6. Differences in fairness definitions or assumptions
> 7. Longer composite actions in AI models
> 8. Unusually technical fairness definitions
> 9. AI overspecialization, such as hard-coded literals that would typically be parameterized
>
> The occurrence of these differences across systems and LLMs is summarized in the tables below:
> | Basic Modeling Agent | Claude-Sonnet-4 | GPT-5            | Gemini-2.5-Pro | DeepSeek-R1   |
> |----------------------|-----------------|------------------|----------------|---------------|
> | Asterinas Spin       | 1, 3            | 1, 5, 8          | 1, 5, 8        | 1, 5, 8       |
> | Asterinas Mutex      | 1, 5            | 1, 5             | 1, 5, 9        | 5             |
> | Asterinas Rwmutex    | 1,5             | 1, 5             | 1, 5           | 1, 5          |
> | Etcd Raft            | 1, 2, 4, 5, 7   | 1, 2, 4, 5       | 1, 2, 4, 5, 7  | 1, 2, 4, 5, 7 |
> | Redis Raft           | 1, 4            | 1, 4, 7          | 1, 4, 7        | 1, 4          |
> | Xline CURP           | 1, 2, 4, 5, 6   | 1, 2, 3, 4, 5, 6 | 1, 2, 4, 5, 6  | 1, 2, 4, 5, 6 |
> | PGo dqueue           | 1, 5            | 1, 5             | 1, 5           | 1, 5, 7       |
> | PGo locksvc          | 1, 5            | 1, 5             | 1, 5           | 5, 6, 7, 8    |
> | PGo raftkvs          | 1, 5, 7         | 1, 5, 7, 8       | 1, 5, 8        | 1, 5          |
>
> | Code Translation Agent | Claude-Sonnet-4  | GPT-5         | Gemini-2.5-Pro   | DeepSeek-R1      |
> |------------------------|------------------|---------------|------------------|------------------|
> | Asterinas Spin         | 1, 2, 3, 5, 6    | 1, 3, 5, 6, 8 | 1, 2, 3, 5, 6    | 1, 2, 3, 5, 6, 8 |
> | Asterinas Mutex        | 1, 2, 5, 8       | 1, 2, 5       | 1, 2, 5          | 1, 2, 5, 8       |
> | Asterinas Rwmutex      | 1, 5, 6          | 1, 3, 5, 6, 8 | 1, 3, 5, 6       | 1, 5, 6, 8       |
> | Etcd Raft              | 1, 2, 4, 5, 6, 8 | 1, 2, 4, 5, 6 | 1, 2, 4, 5, 6, 7 | 1, 2, 4, 5, 6, 7 |
> | Redis Raft             | 1, 3, 4, 5       | 1, 3, 4, 5    | 1, 3, 4, 5       | 1, 2, 4, 5, 6    |
> | Xline CURP             | 1, 4, 5, 6       | 4, 5, 6       | 1, 2, 4, 5, 6    | 1, 4, 5, 6       |
> | PGo dqueue             | 1, 2, 3, 5, 6    | 1, 5, 6       | 1, 5, 6          | 1, 3, 5, 6       |
> | PGo locksvc            | 1, 5             | 1, 5, 8       | 1, 5             | 1, 5, 6, 8       |
> | PGo raftkvs            | 1, 5, 7          | 1, 5, 7, 8    | 1, 5, 6, 7, 8    | 1, 5, 6, 7       |
>
> **2. Bug reproduction from previous system versions**
>
> Without any hints about the bugs, AI-generated models successfully reproduced several hard-to-find bugs across multiple systems, demonstrating that these models can be practically useful for human engineers in partial correctness checking.
>
> | Reproduced bugs   |                        link                        |
> |-------------------|--------------------------------------------------|
> | Etcd Raft         | https://github.com/etcd-io/etcd/pull/10998         |
> | Xline Curp        |    https://github.com/xline-kv/Xline/issues/402    |
> | Redis Raft        |  https://github.com/RedisLabs/redisraft/issues/19  |
> | Asterinas mutex   |  https://github.com/asterinas/asterinas/pull/1279  |
> | Asterinas rwmutex | https://github.com/asterinas/asterinas/issues/1303 |
>
> These bugs were discovered by having AI agents generate TLA+ models from earlier versions of the system code and then using TLC model checking to identify the issues.

---

> > ### Author Response · Authors · 2025-11-21
> > **We hope to clarify several misunderstandings in the weakness comments.**
> >
> > **> C1: do not directly measure whether AI truly captures the semantic intent or design rationale of the system**
> >
> > Thanks for the question. You are right that system modeling may not directly reflect how AI truly captures semantic intent or design rationale. Our experience (the code translation agent) shows that one can write reasonable formal models by translating source code to specifications in a bottom-up manner. We will make this clear in the paper.
> >
> > The task – modeling systems in TLA+ – comes from a true desire for automated system verification. It can be done in different ways. Ideally, LLMs should truly capture the semantic intent or system design rationale. SysMoBench evaluates this using the modeling agent – when the model-level and the implementation-level behavior are equivalent, we consider the model captures the semantic intent of the system implementation.
> >
> > ---
> >
> > **> C2: Some system code (e.g., Etcd Raft) may already appear in LLM training data, possibly affecting fairness; the paper lacks a clear anti-contamination discussion**
> >
> > Thanks for raising this point. We did not make it clear – in fact, we have considered the data contamination problem.
> >
> > First, this is intended to have system code in LLM training data – we want LLMs to learn the target system code. It is a similar process as how humans write formal models – they first spend years to learn the details of system code and then write the formal model for it. A recent report shows that PhD students take 5 years to learn system code and write TLA+ models. Note that this is different from coding benchmarks in that we ask LLMs/agents to write the code.
> >
> > Second and more importantly, few systems in Table 1 have the TLA+ system models in their open-source repositories. The TLA+ specifications of Asterinas Spinlock/Mutex/Rwmutex are never released. Redis Raft and Xline CURP do not have any TLA+ specifications. Etcd Raft and PGo systems do have TLA+ specifications in their repositories. However, those specifications are protocol specifications instead of system specifications. The former describes high-level abstractions at the protocol level, which does not describe important details of the system implementations.
> >
> > In fact, the only whole-system specification we are aware of is ZooKeeper’s ZAB implementation (Ouyang et al., 2025). We did not include it in the benchmark.
> >
> > Our goal is in fact to leverage LLMs/agents to write effective TLA+ specifications for all important, safety-critical software systems in the wild. If this goal can be realized, it can enable a number of exciting testing and verification techniques to automatically improve the quality and reliability of these systems.
> >
> > ---
> >
> > **> C3: Limited scalability despite claims of automation**
> >
> > By “automation”, we meant that the evaluation is automated without human intervention.
> >
> > We agree that nine systems do not look large. In fact, this reflects the key challenges of developing system-level AI benchmarks beyond code-snippet-level benchmarks – despite many coding benchmarks, there are few system benchmarks, because system artifacts are large, complex, and hard to prepare, especially by benchmark developers who were not the developers of the systems. As a result, system intelligence of existing LLMs is weak.
> >
> > We believe that the only way to make progress is to gradually develop system-level benchmarks through a community-based approach, where the system developers add their own system artifacts into the benchmark. SysMoBench shows such a feasibility. In fact, the system artifacts we added in the benchmark were those we ourselves have been working on for years. That’s the reason that it only takes us 3-4 days to add one artifact. For systems that we never worked on or are unfamiliar with, it will take much longer.
> >
> > Note that this is the same process as many popular benchmarks were developed. For example, the original SWE-bench only uses 12 open-source project repositories and it has been growing ever since.

---

> ### Comment · Reviewer_K1Um · 2025-11-22
>
> I am satisfied with the authors’ rebuttal, and I believe that most of my earlier concerns have been adequately addressed. Regarding the discussion on Alloy and PAT with LLMs, I have recently read works such as [1] and [2]. Interestingly, their conclusions appear to diverge. I am wondering whether this discrepancy primarily arises from differences in the datasets or whether current LLMs are not sufficiently trained on them as what your said in the rebuttal.
>
> [1] Zuo X, Zhang Y, Wang H, et al. PAT-Agent: Autoformalization for Model Checking. International Conference on Automated Software Engineering (2025).
> [2] Alhanahnah, M., Rashedul Hasan, M., Xu, L. et al. An empirical evaluation of pre-trained large language models for repairing declarative formal specifications. Empirical Software Engineering 30, 149 (2025). https://doi.org/10.1007/s10664-025-10687-1

---

> > ### Author Response · Authors · 2025-11-23
> > **We have read the papers and tried out the research artifacts, based on which we present our comparative analysis.**
> >
> > Thank you for sharing the two papers. We have carefully read them and also tried out the research artifacts of the two papers, based on which we present the comparative analysis.
> >
> > **1. Reasons of Discrepancy**
> >
> >  The discrepancy comes from two main sources: the nature of the tasks and their complexity.
> >
> > (1) The tasks for LLMs/agents to solve are very different.
> >
> > | Benchmark | Task |
> > | -------- | -------- |
> > | SysMoBench (our work) | Generating formal models for _real-world_ software systems from their code |
> > | PAT-Agent [1] | Generating formal models in PAT from natural language descriptions |
> > | Alloy-APR [2] | Repairing a given formal model in Alloy with injected errors |
> >
> > **We believe that repairing a formal model is easier than generating a model from scratch.**
> >
> > (2) The complexity and scale of the tasks are very different. SysMoBench's tasks (generating formal models from system code) are much more complex than the tasks in PAT-Agent and Alloy-APR.
> >
> > Since the tasks are very different, it is hard to directly compare the complexity of the three benchmarks. Therefore, we compare the complexity of the generated formal models, as follows.
> >
> > | Bench   | LoC (smallest model)   | LoC (largest model)  | LoC (median) | Task |
> > | ---- | ---- | ---- | ---- | ----- |
> > | SysMoBench  | 75 (dqueue)  | 508 (raftkvs)  | 219            | Generation |
> > | PAT-Agent [1]         | 16 (sliding_game) | 142 (tesla)    | 45             | Generation |
> > | Alloy-APR [2]: ARepair | 15 (student15_0)  | 99 (farmer1_0)   | 50  | Repair |
> > | Alloy-APR [2]: Alloy4Fun | 1 (trash_inv10_21_0_False) | 234 (classroom_inv15_59_1_False) | 21.0           | Repair |
> >
> > **Compared with the generation task in PAT-Agent, seven out of the nine models we expect LLMs/agents to generate in SysMoBench is larger than the largest models in the PAT-Agent paper.**
> >
> > Specifically, the PAT-Agent tasks are small samples such as river-crossing puzzles and restaurant workflows, not real-world software systems.
> >
> > Alloy-APR's tasks are similar, which comes from ARepair and Alloy4Fun. *Neither of them uses real-world system artifacts.*
> >
> > **2. Reproducing Alloy-APR and PAT-Agent Results**
> >
> > To verify that our setup is correct and consistent with the two papers, we tried to reproduce the results of the two papers. We report the results of Alloy-APR first, as we cannot reproduce some of the results of PAT-Agent.
> >
> > (1) **Alloy-APR [2]**
> >
> > The artifact of Alloy-APR is well prepared. We reproduced its setup and ran it with our models used in SysMoBench.
> >
> > The following table presents the results of ARepair which contains artificially constructed small problems involving simple quantifiers or set operations. It asks the LLMs/agents to repair injected errors.
> >
> > We can see that Claude and GPT-5 already outperform the Best Results in the paper, which suggests that existing LLMs indeed can solve these tasks — the high scores in the paper are largely due to the fact that the task itself is relatively simple.
> >
> > | Model              | Correct Items | Ratio  |
> > | ----- | ----- | ----- |
> > | Claude             | 38 / 38        | 100%   |
> > | GPT-5              | 30 / 38        | 78.9%  |
> > | Gemini             | 14 / 38        | 36.8%  |
> > | DeepSeek           | 5 / 38         | 13.2%  |
> > | Best Result in [2] | 28 / 38        | 73.7%  |
> >
> > We have not run Alloy4Fun because the task is similar, and the paper reports similar results.
> >
> > Clearly, while LLMs/agents can repair 1-3 lines of errors in small Alloy models, it can hardly generate syntax-correct Alloy models from complex system code.
> >
> > (2) **PAT-Agent [1]**
> >
> > We failed to reproduce the results of PAT-Agent – the quality of its research artifact is low. We tried 6 cases in the artifacts: 4 cannot compile; 1 had an assertion error, and only 1 passed. Note that we do not intend to criticize the artifacts (it could be that the PAT-Agent authors uploaded the wrong code).
> >
> > Since we cannot reproduce the results, we directly run the NoPlanning workflow in PAT-Agent (after fixing a few errors in the artifacts) using the LLMs used in SysMoBench. This NoPlanning workflow simply calls the LLM to generate a PAT model and then iteratively fixes it, similar to our basic modeling agent.
> >
> > The results are as follows:
> >
> > | Model | CSR        | FPR        | APR        |
> > | ----- | ----- | ----- | ----- |
> > | Claude-4.0-sonnet | 0.8462 | 0.8077     | 0.8734     |
> > | Gemini-2.5-pro | 0.8462 | 0.6538     | 0.7484     |
> > | GPT-5 | 0.8462    | 0.6923    | 0.7638    |
> > | DeepSeek-R1 | 0.5769    | 0.5000 | 0.5490    |
> >
> > The results are consistent with the paper. Similar to the observation of [2], LLMs can solve these simple tasks to some extent. However, its ability to generate formal models for real-world software systems is much weaker.
> >
> > **3. Summary**
> >
> > The comparative analysis clearly shows the discrepancy between the tasks of generating formal models from large, complex software system code, vs. repairing small models [2] and generating simple models from natural language descriptions.

---

> ### Comment · Reviewer_K1Um · 2025-11-24
>
> I am satisfied with the comprehensive analysis, and I agree that the differing conclusions are most likely attributable to differences in the datasets used. It may be beneficial to include a comparison of these datasets in the related work section to better highlight the uniqueness and complexity of your own dataset. Additionally, you may emphasize that your dataset reveals additional limitations of current AI models in model checking (and also extensible). Based on this, I have increased my score.

---

> > ### Author Response · Authors · 2025-11-24
> > **Thank you for being so engaging and open-minded! We will follow your suggestions!**
> >
> > Thank you for being open-minded and being so engaging in the discussion.
> >
> > We will include the comparison of these datasets in the paper to highlight the uniqueness and complexity of SysMoBench, as well as how it reveals additional limitations of current AI models in formal system modeling. Thank you for the suggestions!
> >
> > Let us know if you have any more feedback or any questions we can answer.
> >
> > Thanks again!

---

### Official Review · Reviewer_Sb7c · 2025-10-29

**Soundness:** 3
**Presentation:** 2
**Contribution:** 2
**Rating:** 2
**Confidence:** 3

**Summary:**

This paper introduces SysMoBench, which is a benchmark framework for evaluating the outputs of large language models using a formal specification language called TLA^{+}. The authors developed nine benchmarks and evaluated the types of syntax and runtime errors, as well as violations of safety and liveness specifications. The authors evaluate the frontier models by expressing them using TLA^{+} language outputs.

**Strengths:**

The authors develop a benchmarking tool to evaluate LLM outputs with tools from formal methods, and using a formal specification language. The benchmark also evaluates several types of errors, such as syntax and runtime errors, as well as violations of safety and liveness specifications.

**Weaknesses:**

I think the benchmarks in the paper mainly demonstrate the lack of capability of outputting TLA^{+} models for these models, which is mentioned in the "Analysis on LLMs". I think that the reason Claude-sonnet is overperforming is mainly because it's more adept at creating outputs with TLA^{+} syntax, but not necessarily having a superior ability to reason and generate real-world system artifacts.

**Questions:**

The authors listed a number of benchmarks, such as MMLU, ARC, and HELM, but I think there's a lack of motivation behind using TLA^{+}. There are also some comparisons missing relevant to structured reasoning, such as EvalScope and Agent-SafetyBench. I would like to understand why using TLA^{+} is a much better alternative than these benchmarks, considering that it's not a very common specification language, such as linear temporal logic. The authors should also spend some time on defining the basic properties of TLA^{+}.

I think it's also unclear how the benchmark tasks were generated, and how this tool can be extended to use different benchmarks. The authors mention this in a paragraph, but I don't see the point of adding their subjective experience without an extensive introduction in the appendix.

---

> ### Author Response · Authors · 2025-11-21
> **We would like to clarify several misunderstandings.**
>
> Thanks for the detailed review. We appreciate it.
>
> Reading the review, it appears that there are clear misunderstandings of the work – we as authors should have done a better job of providing more background information on the tasks and specification languages.
>
> We have improved the draft and uploaded the new version. Please check it out.
>
> In the rebuttal, we hope to clarify a few misunderstandings. We hope to emphasize that our work targets an important real-world problem – formally model critical computing systems as the foundation of formal verification of these systems – instead of attempting to develop a generic framework for reasoning and problem-solving.
>
> ---
>
> **> Q1: Why using TLA+ is a much better alternative considering that it's not a very common specification language**
>
> TLA+ is the primary and _de facto_ formal specification language for distributed and concurrent systems in practice. It is widely used by software companies like Amazon, Microsoft, Nvidia, Google, Oracle, etc (see TLA+ Foundation (2025)) to check and verify critical infrastructure systems such as distributed consensus systems (e.g., Etcd and ZooKeeper), confidential consortium frameworks (Howard et al., 2025), databases (e.g., CosmosDB and MongoDB), OS kernel synchronization (Tang et al., 2025), and cache coherence (Beers, 2008). It is not unfair to say that there is no better alternative than TLA+ with respect to formally modeling real-world computing systems.
>
> ---
>
> **> Q2: Why not linear temporal logic?**
>
> TLA+ is the most widely used specification language for TLA, which _includes_ and _extends_ standard linear temporal logic (LTL).
>
> ---
>
> **> Q3: Comparison with MMLU, ARC, and HELM**
>
> We have cited and discussed these benchmarks in the Related Work. In general, SysMoBench focuses on the specific task of formally modeling large, complex computing systems, which is the foundation of formal system verification in practice. It does not intend to evaluate generic reasoning or problem-solving capabilities like MMLU, ARC, and HELM.
>
> ---
>
> **> Q4: Comparisons missing relevant to structured reasoning, such as EvalScope and Agent-SafetyBench**
>
> We have discussed these two benchmarks in the updated version. The three benchmarks are fundamentally different:
> * SysMoBench focuses on formal modeling of distributed and concurrent computing systems.
> * Agent-SafetyBench is a benchmark on safety of AI agents; SysMoBench does not focus on AI agents, but critical computing infrastructures like operating and distributed systems.
> * EvalScope is a framework for benchmarking large models across many dimensions.
>
> In summary, SysMoBench does not target the safety of AI agents. And, it is not covered by any benchmarks in EvalScope.
>
> ---
>
> **> Q5: How the benchmark tasks were generated?**
>
> The tasks are designed based on the fundamental requirements of a formal system model:
> * The model must be encoded with correct syntax in specification language
> * The model must be able to run without errors during model checking
> * The model must conform to the actual system code
> * The model must satisfy the correctness properties of the target system
>
> The tasks are not from subjective experience, but are the basic building blocks required by any formal verification techniques.
>
> ---
>
> **> Q6: How this tool can be extended to use different benchmarks?**
>
> We believe that you meant the extensibility to other formal method tools than TLA+. The SysMoBench framework is indeed made general to other specification languages.  During this rebuttal period, we added the support for Alloy and PAT as suggested by Reviewer K1Um. The implementation details can be found in Appendix B.1 and B.2.
>
> We used the basic modeling agent to generate the system model in PAT and Alloy. The AI-generated Alloy and PAT models are rather poor – only GPT-5 can generate an Alloy model that passes the syntax correctness check, after many attempts. We believe that the weak model capabilities using PAT and Alloy are mostly because Alloy and PAT are much less popular than TLA+ in real-world systems; so LLMs are not extensively trained on them. This also justify the use of TLA+ as the specification language of choice.
>
> ---
>
> **> Q7: Claude-sonnet is overperforming is mainly coz it's more adapt at creating outputs with TLA+ syntax?**
>
> This is a reasonable observation. Since only syntax-valid models proceed to subsequent evaluation, the ability of passing syntax checking more reliably indeed means an advantage.
>
> It’s worth pointing out that we shall give more credit to Claude-Sonnet-4 beyond its ability of writing syntax-correct TLA+. SysMoBench decomposes the evaluation into four independent metrics, which separate syntactic correctness from reasoning about system behavior. As shown in our updated appendix tables (Tables 9-17), for models that pass syntax checks (e.g., Basic Modeling in Table 11), Claude-Sonnet-4 still achieves higher scores on runtime, conformance, and invariant metrics.

---

### Official Review · Reviewer_f1Je · 2025-10-31

**Soundness:** 3
**Presentation:** 3
**Contribution:** 4
**Rating:** 8
**Confidence:** 2

**Summary:**

This paper presents a new benchmark, SysMoBench, for assessing the ability of AI systems to generate valid formal models of complex real-world systems. The main contributions of the paper are threefold:
1. The authors properly define key metrics to assess the quality of the AI generated system models (based on syntax correctness, runtime correctness, conformance to the system implementation and invariant correctness)
2. The authors also clearly lay out an effective fully automated approach to compute those metrics
3. An empirical evaluation that shows the limitation of state-of-the-art LLMs in generating system models for real-world complex systems.

**Strengths:**

1. **Relevance**
The paper addresses a critical challenge: evaluating the capability of AI systems to move beyond basic code generation and comprehension, toward a deeper understanding and modeling of complex real-world systems.

2. **Significance and Novelty**
To the best of my knowledge, the proposed approach is novel in its problem formulation, the techniques used to fully automate the quality assessment of AI-generated models, and its focus on real-world complex systems. The introduction of this new benchmark holds strong potential for significant impact, as it provides a vital resource that could accelerate progress in AI-driven modeling of complex systems.

3. **Soundness and Experimental Evaluation**
The experimental results validate the effectiveness and robustness of the key LLM-assisted components within the automated evaluation pipeline. They also highlight that current state-of-the-art LLMs still face challenges in accurately modeling complex real-world systems, which underscores the importance of this benchmark in advancing research on AI modeling of intricate software systems.

**Weaknesses:**

I could not find any major problem with the paper.

**Questions:**

The authors write in section 3.2.4:
>"In principle, if a system model fully conforms to code, violations of these invariants would indicate bugs in system code; in practice, few
AI-generated models achieved fine-grained conformance."

Let's assume that, with the help of this new benchmark, state-of-the-art LLMs improve significantly at complex system modeling tasks. What approach would you recommend to determine whether a violation of an invariant stems from a rare LLM modeling error or from an actual bug in the system code?

---

> ### Author Response · Authors · 2025-11-21
> **Thank you for the encouragement on the work!**
>
> Thanks for the insightful review. We really appreciate the feedback! Building a benchmark with real-world system artifacts takes a great amount of effort – thank you for the encouragement on our work!
>
> We answer the question as follows.
>
> --------------------------------------------------------------------------------------------------------------------------------------------
>
> **> Q1: Let's assume that, with the help of this new benchmark, state-of-the-art LLMs improve significantly at complex system modeling tasks. What approach would you recommend to determine whether a violation of an invariant stems from a rare LLM modeling error or from an actual bug in the system code?**
>
> This is a great question! In fact, from a system modeling perspective, conformance is a sufficient metric. If a system model conforms to the system code, then violations of invariants indicate bugs in the code. State-of-the-art model-checking-driven bug finding tools (such as Ouyang et al., 2025 and Tang et al., 2025) follow this principle – they first ensure 100% conformance and then check the model against invariants.
>
> We use invariants as a metric to measure the quality of the TLA+ models mainly because existing LLMs' system modeling abilities are still poor, as reflected in the low conformance metrics. Therefore, we add invariants to reflect another quality dimension and measure whether AI has some concepts of correctness properties such as safety and liveness.
>
> In the future, if LLMs improve significantly and can generate models with high conformance, we would consider turning invariants into a utility metric, assessing whether the generated TLA+ model can find bugs in the code.

---

### Author Response · Authors · 2025-12-03
**Summary of the Final Version of the Paper**

Dear Reviewers and Area Chairs,

We have uploaded the final version of the paper (see OpenReview link: [PDF](https://openreview.net/pdf?id=SAeaTz8YoM)).

The revised content is marked in the blue color (🔵).

The version addresses all the review comments, including:
* Liveness violation analysis (Section I.1)
* Support two new specification languages: PAT and Alloy (Section 3.3)
* Evaluation results of PAT and Alloy in SysMoBench (Section B)
* Qualitative evaluation of AI-generated models (Section C)
* Comparison with human-written models (Section C.1)
* Bug reproduction (Section C.2)
* Articulating and clarifying the choice of using TLA+ as the specification language (Section 2)
* Clarify that TLA+ includes and extends standard linear temporal logic (LTL) (Section 2)
* Added two more systems, Apache ZooKeeper and Ringbuffer, in SysMoBench (Section I.2)
* Clarify that SysMoBench does not require TLA+ models to be structurally equivalent to the system code and does not enforce unnecessary details like choices of data structures. SysMoBench requires behavioral conformance. (Section 3.1)
* Clarification on our approach to prevent training data contamination (Section 4).
* Many revisions with requested details based on Reviewer RwMC’s comments (across the paper).

Please let us know if there are any questions you have or anything we can clarify or provide.

Thank you!

---

### Meta-Review · Area_Chair_7Rtt · 2026-01-05

**Summary:**

The paper proposes a new benchmark for evaluating AI’s ability to generate formal models in TLA+, a language used to specify large and complex computer systems. All reviewers consider the benchmark novel and the evaluation methodology generally valid.

Two reviewers raise concerns about how well the evaluation metrics reflect true AI capability. Reviewer 3 notes the lack of comparison with human-written baselines and raises the possibility of data contamination. Reviewer 4 points out that some evaluation steps, such as checking specification equivalence, are difficult to score automatically.

The authors’ rebuttal provides clarifications and additional results that address these concerns. The new qualitative comparison with human-written models and the clarification of data sources address Reviewer 3’s issues, while the added details on the evaluation procedure are likely to mitigate Reviewer 4’s concerns.

Overall, the paper presents a novel benchmark and reasonable evaluation metrics. Given that Reviewer 3 has updated their score to positive and Reviewer 4’s main concerns appear to be addressed, all remaining reviewers give positive scores, and the recommendation is accept.

The review from Reviewer 2 (Sb7c) is of low quality and is therefore excluded from the above discussion.

**Reviewer Concerns:**

Reviewer 3’s concerns regarding semantic understanding, human baselines, and data contamination have been addressed.

Reviewer 4’s concern about the difficulty of automatic evaluation is likely to be addressed.

Reviewers 3 and 4’s concern about the limited scale of the benchmark is not addressed; the authors acknowledge this as a limitation in the rebuttal.

**Reviewer Scores:**

Reviewer 3 has raised the score.

Reviewer 4 might raise the score, given that the major concern about the evaluation metrics is likely clarified in the rebuttal.

---

### Decision · Program_Chairs · 2026-01-26

Accept (Poster)